# Loss of Parkinson's disease-associated protein CHCHD2 affects mitochondrial crista structure and destabilizes cytochrome c

Hongrui Meng[1,*], Chikara Yamashita[2,*], Kahori Shiba-Fukushima[3], Tsuyoshi Inoshita[3], Manabu Funayama[1], Shigeto Sato[2], Tomohisa Hatta[4], Tohru Natsume[4], Masataka Umitsu[5], Junichi Takagi[5], Yuzuru Imai[2,6] & Nobutaka Hattori[1,2,3,6]

Mutations in *CHCHD2* have been identified in some Parkinson's disease (PD) cases. To understand the physiological and pathological roles of CHCHD2, we manipulated the expression of CHCHD2 in *Drosophila* and mammalian cells. The loss of CHCHD2 in *Drosophila* causes abnormal matrix structures and impaired oxygen respiration in mitochondria, leading to oxidative stress, dopaminergic neuron loss and motor dysfunction with age. These PD-associated phenotypes are rescued by the overexpression of the translation inhibitor 4E-BP and by the introduction of human CHCHD2 but not its PD-associated mutants. CHCHD2 is upregulated by various mitochondrial stresses, including the destabilization of mitochondrial genomes and unfolded protein stress, in *Drosophila*. CHCHD2 binds to cytochrome *c* along with a member of the Bax inhibitor-1 superfamily, MICS1, and modulated cell death signalling, suggesting that CHCHD2 dynamically regulates the functions of cytochrome *c* in both oxidative phosphorylation and cell death in response to mitochondrial stress.

[1] Research Institute for Diseases of Old Age, Juntendo University Graduate School of Medicine, Tokyo 113-8421, Japan. [2] Department of Neurology, Juntendo University Graduate School of Medicine, Tokyo 113-8421, Japan. [3] Department of Treatment and Research in Multiple Sclerosis and Neuro-intractable Disease, Juntendo University Graduate School of Medicine, Tokyo 113-8421, Japan. [4] Molecular Profiling Research Center for Drug Discovery, National Institute of Advanced Industrial Science and Technology, Tokyo 135-0064, Japan. [5] Laboratory of Protein Synthesis and Expression, Institute for Protein Research, Osaka University, Osaka 565-0871, Japan. [6] Department of Research for Parkinson's Disease, Juntendo University Graduate School of Medicine, Tokyo 113-8421, Japan. * These authors contributed equally to this work. Correspondence and requests for materials should be addressed to Y.I. (email: yzimai@juntendo.ac.jp) or to N.H. (email: nhattori@juntendo.ac.jp).

Mutations in the *CHCHD2* gene cause an autosomal dominant form of late-onset PD[1]. Several exonic variants, which may affect the protein levels or subcellular localization of CHCHD2, have been associated with Parkinson's disease (PD) and dementia with Lewy bodies, albeit in a study with a limited number of cases[2].

The gene product CHCHD2 contains a mitochondrial targeting sequence in the N-terminus and two cysteine-$x_9$-cysteine (twin $Cx_9C$) motifs at the C-terminus and has been localized to the intermembrane space of the mitochondria[1,3]. Although little is known regarding the physiological and pathological roles of CHCHD2, the close homologue CHCHD10 is believed to regulate crista structure, maintaining the integrity of the mitochondrial respiratory complexes, at the crista junction of the intermembrane space[4]. A study of yeast CHCHD2 ortholog Mic17 indicated that the loss of Mic17 decreased oxygen consumption and altered activities in respiratory complexes III (ubiquinol-cytochrome *c* [Cyt *c*] reductase) and IV (Cyt *c* oxidase) in *Saccharomyces cerevisiae*[5], whereas an oxidative phosphorylation (OXPHOS) computational expression screen using mouse and human microarray data revealed that CHCHD2 is required for the OXPHOS function in mammalian cells[6]. Further functional analysis has indicated that the loss of CHCHD2 destabilizes complex IV, which reduces oxygen consumption and complex IV activity but not complex I (NADH:ubiquinone oxidoreductase) activity[6]. Another study suggested that CHCHD2 positively regulates the function of complex IV through direct binding and transcriptional regulation of the complex IV subunit-4 isoform COX4I2 (ref. 3). It has also been suggested that CHCHD2 inhibits apoptosis through suppression of Cyt *c* release by binding to Bcl-xL[7].

Dysregulation of the mitochondrial respiratory complexes generates reactive oxygen species (ROS), which leads to oxidative stress and affects the survival and function of neurons[8,9]. Insulin/IGF signalling is a key signalling pathway that maintains mitochondrial activity[10], and the downstream components of insulin/IGF signalling, FoxO and the FoxO target eIF4E-binding protein (4E-BP) regulate mitochondrial functions and aging in *Drosophila*[11,12]. Indeed, 4E-BP, which suppresses cap-dependent protein translation through binding to eIF4E, has been shown to have neuroprotective roles against various stresses in *Drosophila* PD models[13–17].

To understand the physiological and pathological roles of CHCHD2 *in vivo*, we established *CHCHD2* mutant flies because fly models harbouring PD genes associated with mitochondrial functions exhibit pronounced mitochondrial phenotypes and have greatly contributed to the understanding of PD gene functions. Here, we report that the loss of CHCHD2 in flies leads to mitochondrial and neuronal phenotypes associated with PD pathology, including increased sensitivity to oxidative stress and loss of dopaminergic (DA) neurons with age. These phenotypes are rescued by 4E-BP and human CHCHD2 but not by CHCHD2 mutants found in PD cases. Our study suggests that mutations of *CHCHD2* have a loss-of-function aspect in PD, exacerbating oxidative stress and cell death signalling.

## Results

**Generation of CHCHD2 loss-of-function flies.** CHCHD2 orthologs are present in various species, including worm, yeast, and plants, and the affected amino acids found in PD cases are mildly conserved among these species (Supplementary Fig. 1a)[5]. We targeted *Drosophila* CHCHD2 (dCHCHD2) and generated hypomorphic *dCHCHD2^H43^* and revertant alleles by imprecise and precise excision, respectively, using the artificial transposon p{EPgy2}EY05234 (Supplementary Fig. 1b,c). The expression of

*dCHCHD2* transcripts in *dCHCHD2^H43^* was reduced to ~8.5% of the level associated with the revertant allele (Supplementary Fig. 1d). Western blot analysis revealed that expression of the dCHCHD2 protein was almost abrogated in *dCHCHD2^H43^* homozygous flies (Fig. 1a and Supplementary Fig. 1e). We also generated a *dCHCHD2* null allele (*dCHCHD2^−/−^*) mutant, which lacks the coding region of *dCHCHD2*, using CRISPR/Cas9 technology and confirmed that the protein expression disappeared (Fig. 1a and Supplementary Fig. 1e–g). Although *dCHCHD2^H43^* and *dCHCHD2^−/−^* flies were grossly normal, emerging from pupae at the expected Mendelian ratio, the mitochondrial morphology in the indirect flight muscles were affected. The structure of mitochondrial cristae became disordered in 14-day-old *dCHCHD2^H43^* homozygous flies, and a 'swirling' phenotype (white arrowheads) and dilatation of matrix spaces were observed (Fig. 1b,d,e). Similar results were obtained with 14-day-old *dCHCHD2^−/−^* flies (Supplementary Movies 1 and 2), and the disintegration of the mitochondrial cristae progressed in 40-day-old flies (Fig. 1c–e). ATP levels in the thorax muscles were mildly reduced with age in *dCHCHD2^H43^* flies, and the phenotype was exacerbated in *dCHCHD2^−/−^* flies because ATP reduction was detected at an earlier stage (Fig. 1f). Mild atrophy of muscles, irregular arrangement of nuclei and an increase in TdT-mediated dUTP nick end labelling (TUNEL) -positive nuclei were observed in 30-day-old *dCHCHD2^−/−^* flies (Fig. 1g), whereas an increase in the number of TUNEL-/tyrosine hydroxylase (TH)-positive neurons was not detected at a single point in time (Supplementary Fig. 1h). However, the loss of *dCHCHD2* led to a reduction of TH signals, suggesting that the functions of dopaminergic neurons declined (Supplementary Fig. 1h). Our histochemical analyses of *dCHCHD2* mutant flies indicate that loss of CHCHD2 affects the maintenance of the mitochondrial crista structure, which is important for the activity of the mitochondrial respiratory complex and for muscle cell survival.

**CHCHD2 loss leads to oxidative stress and neurodegeneration.** Although *dCHCHD2^H43^* adult flies emerged normally and showed normal survival at a younger stage, they exhibited shorter lifespans (Fig. 2a). Oxidative stress is considered a prominent causal factor of PD, and mutations in PD-associated genes that regulate mitochondrial functions have been reported to lead to oxidative stress[18–20]. *dCHCHD2^H43^* adult flies were sensitive to oxidative stress from paraquat (Fig. 2b) and hydrogen peroxide (Fig. 2c), and they exhibited mild sensitivity to starvation (Fig. 2d). ROS production from mitochondria in *dCHCHD2^−/−^* embryonic cells was elevated by 1.77-fold (Fig. 2e,f). Consistent with the elevated ROS production from mitochondria, the protein translation regulator 4E-BP was upregulated at both the transcript and protein levels in the thorax of *dCHCHD2^H43^* flies (Fig. 2g,h), and upregulation of 4E-BP protein was observed in *dCHCHD2^−/−^* flies (Fig. 2h). Increased protein oxidation and a decreased GSH/GSSG ratio was also detected in *dCHCHD2^−/−^* flies (Supplementary Fig. 2a,b). Conversely, the levels of the ROS scavengers SOD1 and DJ-1b were not changed in *dCHCHD2^−/−^* flies (Supplementary Fig. 2c)[21,22]. These results suggest that loss of dCHCHD2 causes chronic oxidative stress, overwhelming ROS scavenging activity in flies[13–17].

We next examined the effects of dCHCHD2 loss on the central nervous system. Consistent with the above evidence that chronic oxidative stress occurs following loss of dCHCHD2, a lipid oxidation marker, 4-hydroxy-2-nonenal (4-HNE), was increased in both whole brain and central DA neurons of *dCHCHD2^H43^* flies (Fig. 2i). Immunoreactivity of 8-hydroxy-2'-deoxyguanosine

(8-OHdG), which is a product of oxidatively damaged DNA formed by hydroxyl radicals, was also increased in the whole brain and the PPL1 and PPM3 DA neuron clusters of $dCHCHD2^{H43}$ flies (Supplementary Fig. 2d). The age-dependent loss of DA neurons was observed in the PPL1 cluster of $dCHCHD2^{H43}$ adult flies even though no changes were observed in the TUNEL assay (Fig. 2j,k), and climbing ability, which is regulated by DA neurons, also declined with age (Fig. 2l)[23]. Taken together, these results indicate that loss of *CHCHD2* produces chronic oxidative stress, which causes aging-dependent neurodegeneration in DA neurons.

**CHCHD2 is upregulated upon mitochondrial stresses.** During the stress assays using *Drosophila*, protein levels of dCHCHD2 were markedly changed following mitochondrial stresses (Fig. 3).

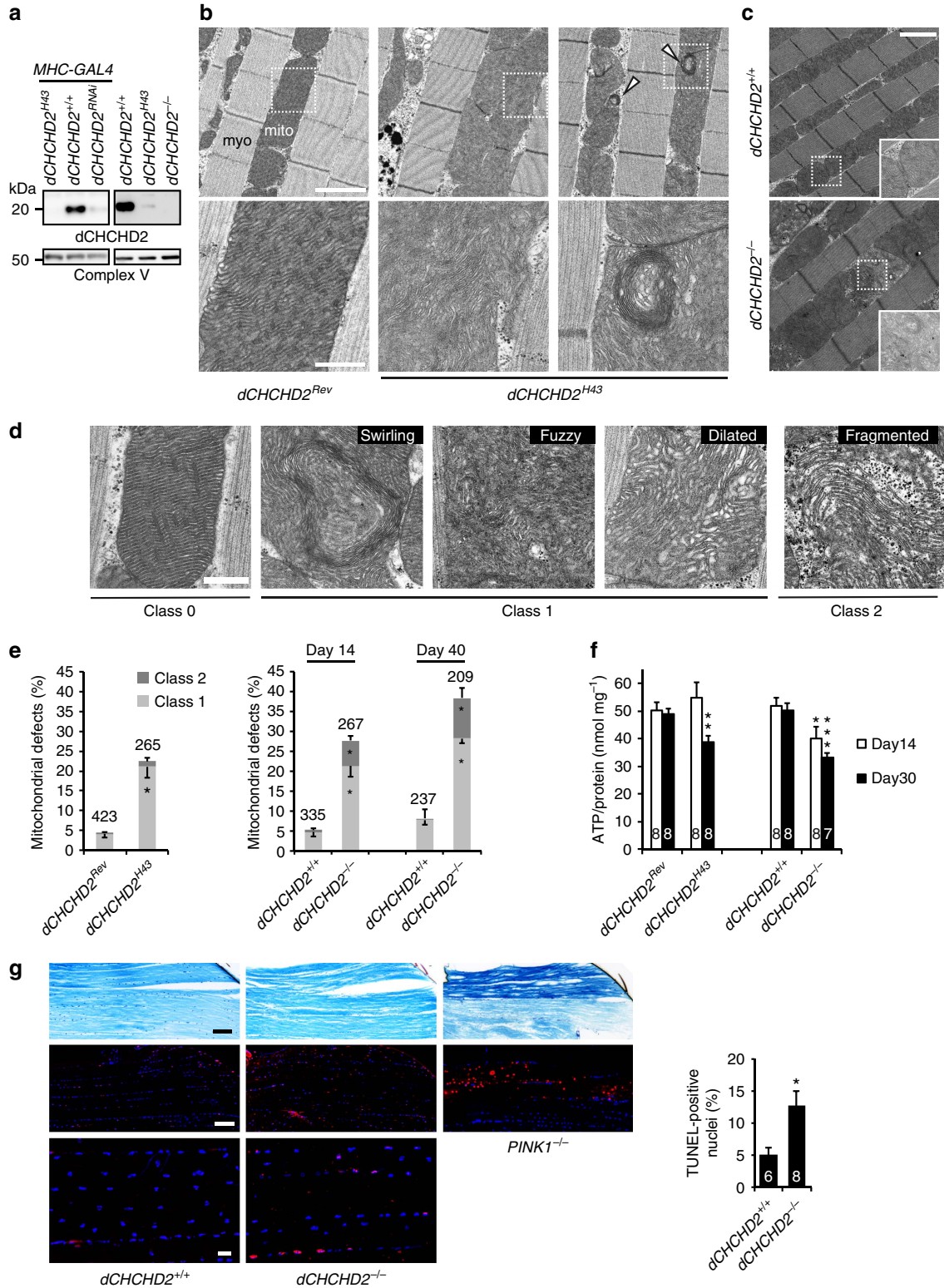

Mitochondrial genomic stress caused by mutations or overexpression of the mitochondrial DNA polymerase γ subunits and mitochondrial unfolded protein stress induced by truncated mitochondrial matrix protein ornithine transcarbamylase (ΔOTC) upregulated dCHCHD2 at both the transcript and protein levels (Fig. 3a,b and Supplementary Fig. 3a)[24,25]. Suppression of the mitochondrial respiratory complex subunits by gene knockdown and loss of PINK1, parkin and DJ-1b also increased or tended to increase dCHCHD2 protein levels (Supplementary Fig. 3b,c), suggesting that CHCHD2 is a mitochondrial stress-responsive protein.

Hyperoxia induces the mitochondrial swirl phenotype and defects in the Cyt c oxidase complex, which is likely to be accompanied by oxidative stress and/or dysregulation of the cristae junctions[26,27]. We thus examined the effects of CHCHD2 loss on hyperoxia. The mitochondrial cristae of the indirect flight muscle became disordered when normal flies were exposed to hyperoxia for 4 days (Fig. 3c,d versus Fig. 1d,e). The phenotype was further exacerbated in dCHCHD2 mutant flies. Consistent with the idea that hyperoxia produces oxidative stress, 4E-BP was upregulated in both normal and mutant flies after hyperoxia exposure (Fig. 3e). Interestingly, dCHCHD2 expression was suppressed by hyperoxia (Fig. 3e). These results suggest that CHCHD2 expression is positively regulated by the reduction in the metabolic status.

**CHCHD2 regulates mitochondrial electron transport.** CHCHD2 has been suggested to be involved in the regulation of OXPHOS in human cultured cells and yeast[6,28], which could be due to the positive regulation of complex IV activity by the binding of CHCHD2 (ref. 3). To determine respiratory ability in the presence or absence of dCHCHD2, we measured the oxygen consumption rate (OCR) of the mitochondria in combination with respiratory complex inhibitors. Although we initially used mitochondria isolated from the flight muscles of adult flies, they did not work well, consistent with a previous report[29]. Then, we employed embryonic cells prepared from $dCHCHD2^{+/+}$ and $dCHCHD2^{-/-}$ flies. Consistent with the finding of lower ATP production in dCHCHD2 mutant flies, a steady OCR was lower in $dCHCHD2^{-/-}$ cells without detectable changes in the OXPHOS proteins (Fig. 4a,b and Supplementary Fig. 2c). Treatment with an inhibitor of ATP synthase, oligomycin, suppressed both OCRs to the same extent (Fig. 4a,b). Subsequent treatment with a proton ionophore, FCCP (carbonyl cyanide 4-[trifluoromethoxy] phenylhydrazone), stimulated maximal OCR, which was lower in $dCHCHD2^{-/-}$ cells (Fig. 4a,b). OCR was largely inhibited by rotenone and antimycin A, indicating that it was derived from mitochondria (Fig. 4a,b). The extracellular acidification rate

(ECAR) was simultaneously measured as an indicator of glycolytic activity, and the basal ECAR values of $CHCHD2^{-/-}$ flies were almost the same, but lower after FCCP treatment, compared with a $CHCHD2^{+/+}$ flies, which suggested that lower OXPHOS activity did not result in activation of glycolysis, at least in an early stage of embryogenesis (Fig. 4c,d). The results from the mitochondrial stress tests imply that $dCHCHD2^{-/-}$ cells are comparable in proton leak to $dCHCHD2^{+/+}$ cells, whereas ATP production and spare respiratory capacity are impaired by the loss of CHCHD2 (Fig. 4e).

The activities of mitochondrial complexes I–IV in mitochondria isolated from the flight muscles of 14-day-old flies were comparable between the two genotypes (Supplementary Fig. 4a). In addition, we could not detect significant binding between CHCHD2 and complex IV (Supplementary Fig. 4b). These results suggest that CHCHD2 maintains a proper electron flow rate from complexes I and II to complex IV, rather than activating complex IV in Drosophila, although we cannot completely exclude the possibility that a very small loss of OXPHOS enzyme activities results in OCR reduction.

**CHCHD2 maintains mitochondrial crista integrity.** A close homologue of CHCHD2, CHCHD10 (identity 58% by ClustalW), is reported to be enriched at the crista junction[4]. We thus searched for the binding partners of CHCHD2 that are involved in the OXPHOS pathway by mass spectrometry analysis. Among them, we focused on an inner mitochondrial membrane protein, MICS1, which has been reported to maintain crista structure and to negatively regulate Cyt c release by direct binding during apoptosis (Supplementary Fig. 5a)[30]. Because MICS1 contains several transmembrane domains, it can be solubilized only in detergent-containing solutions. However, because it was difficult to detect stable binding between human CHCHD2 (hCHCHD2) or Cyt c and MICS1 in detergent-containing solutions, probably due to the unfolding of MICS1 by the removal of the inner mitochondrial membrane, we used a thiol-cleavable crosslinker (dithiobis (succinimidyl propionate); DSP) to detect the binding in their native conformations. Human embryonic kidney (HEK) 293T cells were transiently transfected with plasmids encoding MICS1, hCHCHD2 or Cyt c. We first confirmed the interaction between MICS1 and hCHCHD2 in mitochondria (Supplementary Fig. 5b,c). We then examined whether these three proteins form a tertiary complex in HEK293T cells (Fig. 5a). Cyt c with a FLAG tag was immunoprecipitated with anti-FLAG beads, and co-precipitated MICS1-HA and hCHCHD2-Myc were detected in western blots. Under this condition, an unrelated inner mitochondrial membrane protein, Tim23, was not present in this complex, indicating that DSP specifically links protein complexes.

**Figure 1 | Loss of CHCHD2 impairs mitochondrial function.** (**a**) dCHCHD2 protein in the thorax muscles of normal and mutant flies with the indicated genotypes. Complex V (ATP5a) served as a loading control. Revertant or w- served as a WT allele ($dCHCHD2^{+/+}$), both of which express dCHCHD2 to almost the same levels (Supplementary Fig. 3b). (**b**) TEM images of the indirect flight muscles of 14-day-old adult flies with the indicated genotypes are shown. (upper) Mitochondrion, mito; myofibril, myo. White arrowheads indicate swirling cristae. (lower) Higher-magnification images of boxed regions in upper panels. Scale bars, 2 μm (upper) and 500 nm (lower), respectively. (**c**) TEM images of the indirect flight muscle of 40-day-old adult flies with the indicated genotypes are shown. Higher-magnification images of boxed regions are also inserted. Scale bar, 2 μm. (**d**) Classification used to quantify the frequency of abnormal mitochondria: class 0, normal; class 1, swirling, fuzzy or dilated cristae; class 2, fragmented cristae and loss of electron density. Scale bar, 500 nm. (**e**) Mitochondria with abnormal cristae or degenerating mitochondria were quantified using the scoring system shown in **d**. Mitochondrial defects defined as classes 1 and 2 were counted and are presented as percentages (mean ± s.e.m.). *$P < 0.0001$ versus the same class of age-matched controls. $n = 209$–423 from three independent samples. (**f**) ATP contents of thorax muscle tissues from the indicated genotypes (14-day- and 30-day-old) were measured. ATP contents were normalized to the protein levels (mean ± s.e.m.). *$P < 0.05$, **$P < 0.01$, ***$P < 0.001$ versus age-matched controls. (**g**) Longitudinal sections from flight muscles in the thoraxes of 30-day-old male flies. Toluidine blue staining (upper) and TUNEL (red in middle and lower)-positive nuclei (blue) are shown, and 14-day-old $PINK1^{-/-}$ flies were used as a positive control. The graph shows that TUNEL-positive nuclei were increased in $dCHCHD2^{-/-}$ flies ($P = 0.02$; $n = 6$–8 images from 3 to 4 flies each). Scale bars, 50 μm (upper and middle) and 10 μm (lower), respectively. The numbers of samples analysed are indicated in the graphs (**e**–**g**). TEM, transmission electron microscopy.

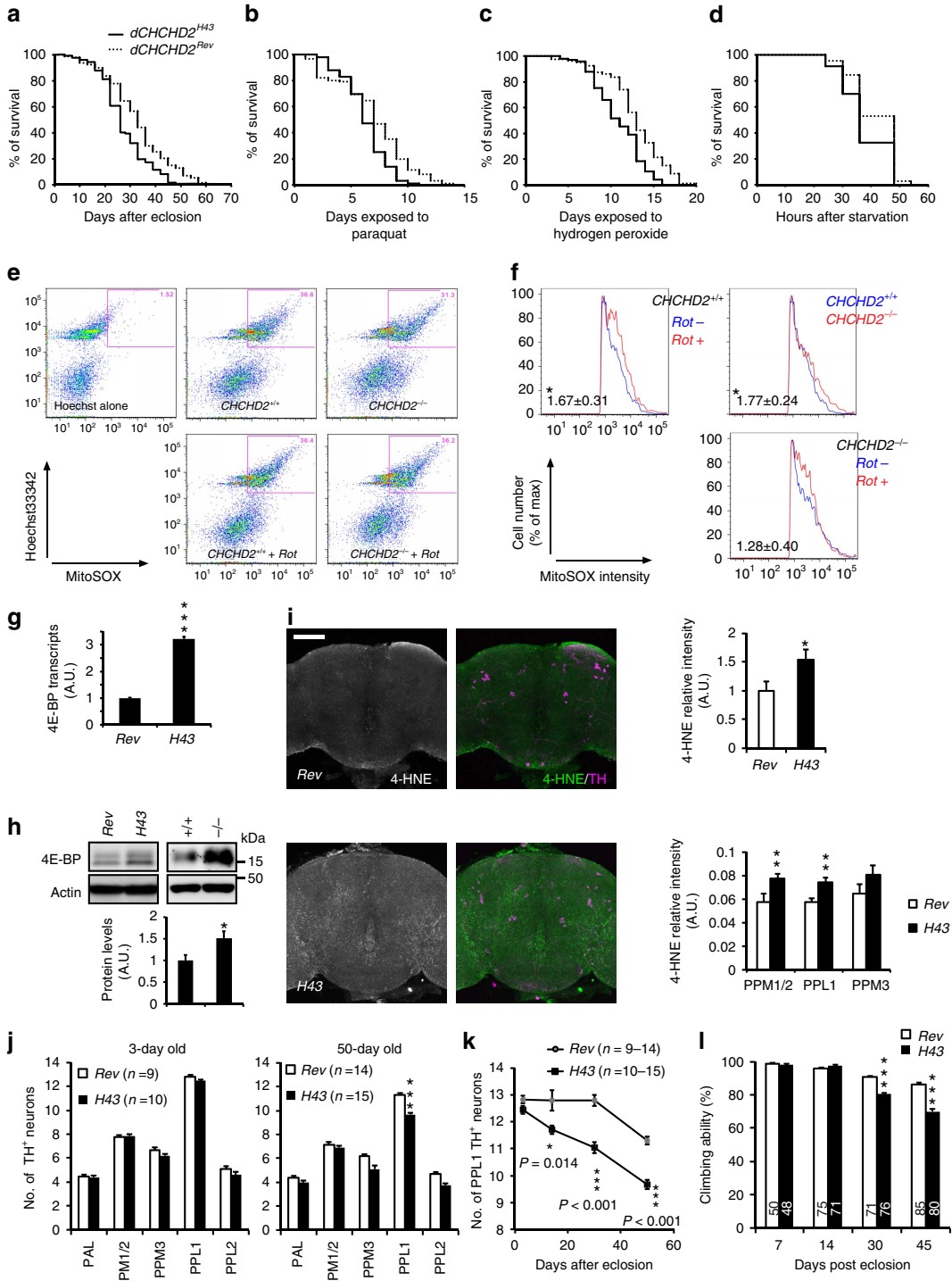

**Figure 2 | dCHCHD2 mutant flies exhibit phenotypes associated with PD.** (**a**) *dCHCHD2*[H43] adult flies showed shortened lifespans compared with *dCHCHD2*[Rev]. $P < 0.00016$ by log-rank test; $n = 122$ (H43) and 127 (Rev) male flies. (**b–d**) *dCHCHD2*[H43] flies are sensitive to 2 mM paraquat treatment (**b**, $P < 0.0030$ by log-rank test; $n = 96$ (H43) and 99 (Rev) males), 0.5% hydrogen peroxide treatment (**c**, $P < 0.000087$; $n = 98$ (H43) and 81 (Rev) males) and starvation (**d**, $P < 0.023$; $n = 117$ (H43) and 111 (Rev) males). (**e,f**) MitoSOX indicates chronic ROS production from mitochondria in *dCHCHD2*[−/−] cells. FACS plots were gated on Hoechst/MitoSOX-positive cells (**e**), and relative MitoSOX intensities are shown as histograms (**f**). Rotenone (Rot, 1 μM for 90 min) served as a positive control. The values indicate the average fold increase ± s.e.m. from three independent experiments. *$P < 0.027$. (**g,h**) 4E-BP is upregulated at both the transcript (**g**) and protein (**h**) levels (mean ± s.e.m., $n = 3$) in *dCHCHD2* mutant flies. *$P = 0.050$, ***$P = 0.000019$. (**i**) Lipid oxidation is exacerbated in *dCHCHD2*[H43] flies. The whole brain tissues of 30-day-old flies were stained with anti-4-HNE and anti-dTH antibodies. Scale bar, 75 μm. Intensity of anti-4-HNE immunoreactive signals in whole brain (upper graph, $n = 5$) and TH-positive regions (lower graph, $n = 39$) were measured and graphed (mean ± s.e.m.). *$P = 0.048$, **$P < 0.009$ versus *Rev*. (**j,k**) The number of PPL1 DA neurons decreases in *dCHCHD2*[H43] flies with age. (**j**) The number of the indicated cluster of DA neurons (mean ± s.e.m.) was counted. ***$P < 0.0001$ versus *Rev*. (**k**) The number of PPL1 DA neurons at 3, 14, 30 and 50 days of age (mean ± s.e.m.). (**l**) The motor defect of *dCHCHD2*[H43] flies with age. Mean ± s.e.m. from 20 trials with 48–85 male flies from three independent experiments. ***$P < 0.0001$ versus age-matched *Rev*. (**g–l**) two-tailed student's *t*-test. FACS, fluorescence-activated cell sorting.

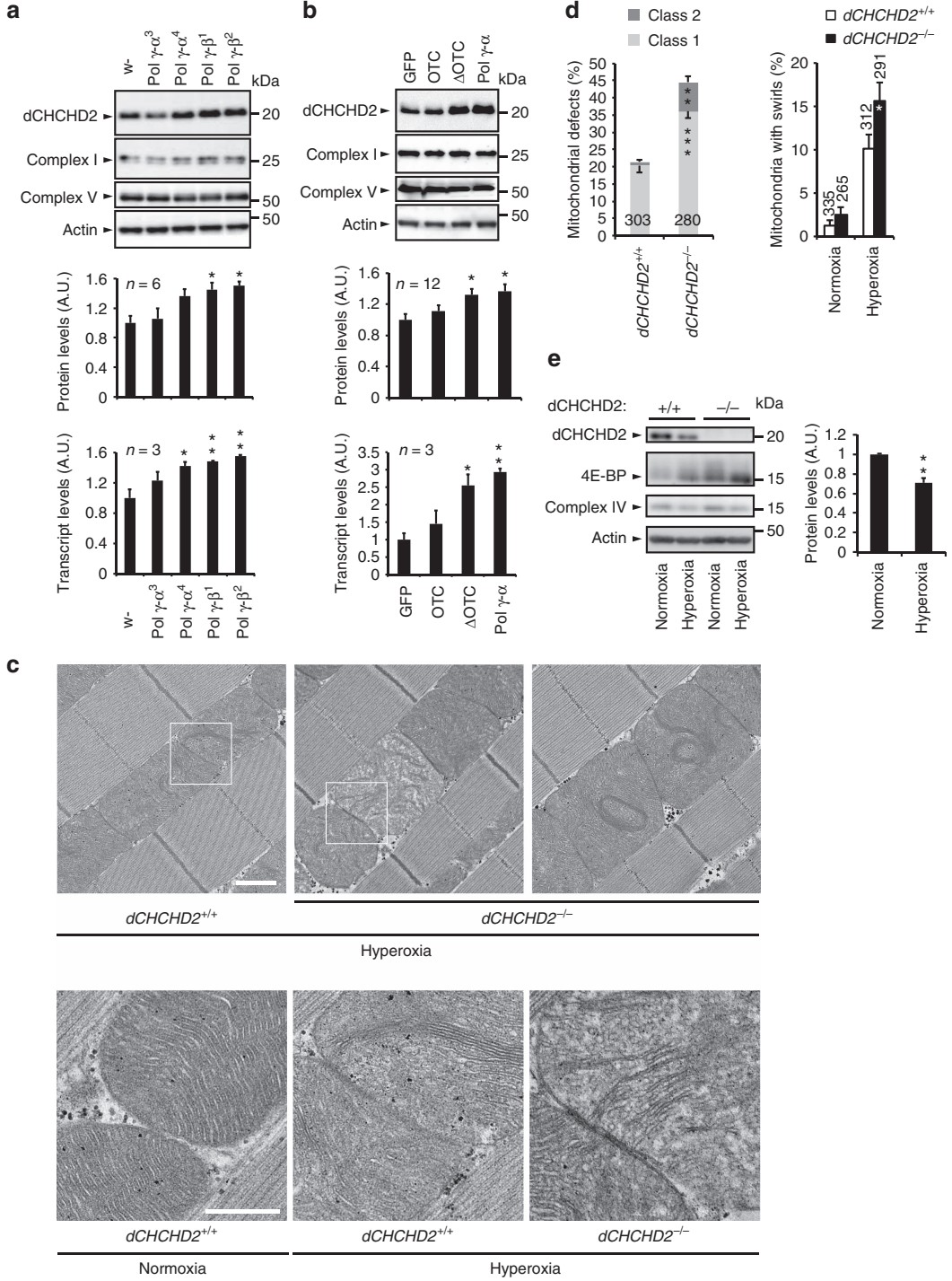

**Figure 3 | CHCHD2 responds to mitochondrial stresses. (a,b)** dCHCHD2 responds to mitochondrial DNA damage and mitochondrial unfolded protein stress. (**a**) Thorax muscle tissues of flies harbouring mutations in the α- or β-subunit of the mitochondrial polymerase γ were subjected to western blot analysis with the indicated antibodies. (**b**) Overexpression of ΔOTC, but not OTC, produced unfolded protein stress in mitochondria, whereas overexpression of the polymerase γ, α subunit destabilized the mitochondrial genome[24]. The indicated UAS-transgenes were driven by *MHC-GAL4*. NDUFS3 (Complex I), ATP5a (Complex V) and actin served as loading controls. *w-* and GFP were used as control lines. The graph represents relative values (mean ± s.e.m.). *$P < 0.05$, **$P < 0.01$ by one-way ANOVA with Tukey–Kramer test. (**c**) The loss of CHCHD2 results in sensitization to hyperoxia. TEM images of the indirect flight muscles of adult flies with the indicated genotypes incubated under 90% $O_2$ for 4 days are shown (upper). Higher-magnification images of boxed regions in upper panels (lower). Normal mitochondria of 14-day-old flies are also shown as a control (*dCHCHD2*[+/+], Normoxia). Scale bars, 1 μm (upper); and 500 nm (lower), respectively. (**d**) Mitochondria with abnormal cristae or degenerating mitochondria in (**c**) were counted as shown in Fig. 1e (mean ± s.e.m.). $n = 265$–335 from three independent samples. *$P = 0.045$, **$P = 0.0002$, ***$P < 0.0001$ versus the same class of *dCHCHD2*[+/+]. (**e**) CHCHD2 is downregulated under hyperoxia. The indicated proteins were analysed using thorax muscle tissues by western blot, and dCHCHD2 levels were graphed (mean ± s.e.m.). **$P = 0.0054$. $n = 5$ from two independent experiments. (**d,e**) two-tailed student's *t*-test. TEM, transmission electron microscopy. ANOVA, analysis of variance.

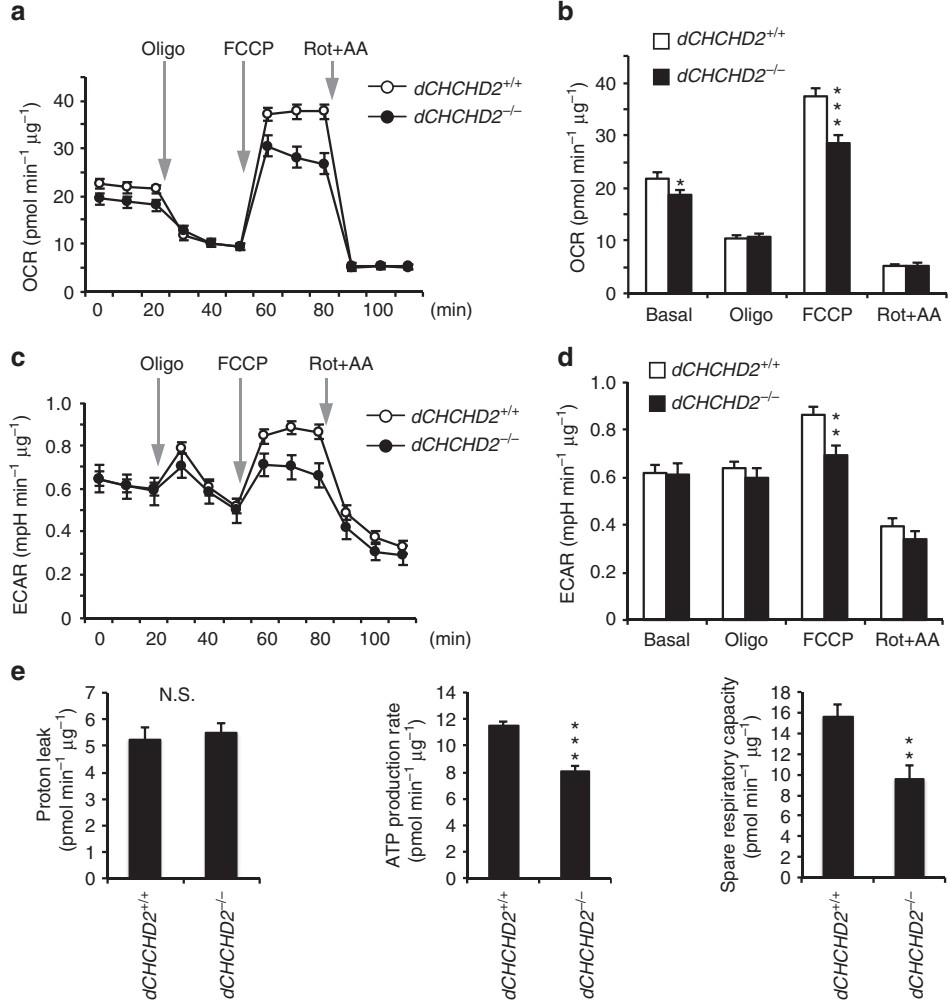

**Figure 4 | CHCHD2 maintains the electron flow in OXPHOS. (a–d)** The function of the mitochondrial electron respiratory complex in *Drosophila* embryonic cells was assessed as the OCR and the ECAR via serial injection of 5 μM oligomycin (Oligo), 1 μM carbonyl cyanide 4-(trifluoromethoxy) phenylhydrazone (FCCP), and 5 μM rotenone plus 5 μM antimycin A (Rot + AA). The mitochondrial content was comparable between the two genotypes, as estimated based on mitochondrial proteins (Supplementary Fig. 2c). The OCR and ECAR values (mean of the three points ± s.e.m.) are graphed in **b** and **d**, respectively. **(e)** Proton leak from the intermembrane space, ATP production, and spare respiratory capacity (mean ± s.e.m.) were calculated using basal OCR and its alterations with the inhibitors against each respiratory complex. **(a–e)** $n = 17$ ($dCHCHD2^{+/+}$) and 19 ($dCHCHD2^{-/-}$) from three independent experiments. $*P = 0.028$, $**P < 0.01$, $***P < 0.001$; N.S., not significant ($P = 0.66$) by two-tailed student's $t$-test.

Importantly, the amounts of co-precipitated MICS1 with Cyt c-FLAG were largely unchanged in the presence or absence of hCHCHD2-Myc and vice versa, suggesting that MICS1 and CHCHD2 form a tertiary complex with Cyt c rather than compete for binding of Cyt c (Fig. 5a). Endogenous interactions of three proteins were also detected (Fig. 5b). CHCHD2 T61I mutation found in PD cases affected the binding to MICS1 (Supplementary Fig. 5d)[1]. Interaction of CHCHD2 with Cyt c was unchanged by PD mutations (Supplementary Fig. 5e).

We stably introduced the genes for wild-type (WT) hCHCHD2 and PD mutants (T61I and R145Q) in $CHCHD2^{-/-}$ mouse embryonic fibroblasts (MEFs) (Supplementary Fig. 6a,b). WT hCHCHD2 but not mutants suppressed Cyt c release (Fig. 5c,d) and caspase activation (Fig. 5e,f) by actinomycin D. More prominent effects were observed regarding Cyt c release caused by peroxide (Supplementary Fig. 6c).

Knockdown of the *Drosophila* MICS1 ortholog dMICS1, the efficiency of which was ~55%, had little effect on mitochondrial morphology in the $dCHCHD2^{Rev}$ genetic background, whereas it dramatically degenerated the mitochondrial matrices in the $dCHCHD2^{H43}$ background (Fig. 5g,i versus Fig. 1b,e and Supplementary Fig. 7a). Ubiquitous overexpression of dMICS1 by *Da-GAL4* at 25 °C results in death at the 1st instar larval stage. Reduced expression levels (*Da-GAL4* at 22 or 18 °C) caused reduced hatching efficiency (~10%), slowed growth and death at the second instar larval stage, whereas the loss of dCHCHD2 did not affect its developmental phenotype and hatching efficiency of dMICS1-overexpressing larvae (Supplementary Fig. 7b). Muscle-specific overexpression of dMICS1 was viable and promoted mitochondrial fragmentation in both genetic backgrounds, showing mild mitochondrial defects to the same extent (Fig. 5h,i versus Fig. 1b,e). Consistent with the morphological defects, knockdown of dMICS1 largely reduced ATP production in $dCHCHD2^{H43}$ flies (Fig. 5j). Importantly, overexpression of dMICS1 rescued the ATP reduction and motor defects observed in $dCHCHD2^{H43}$ flies, which strongly suggested that MICS1 maintains OXPHOS function in cooperation with CHCHD2 (Fig. 5j and Supplementary Fig. 7c).

**Rescue of fly phenotypes by expression of human CHCHD2.** We next examined whether hCHCHD2 rescues the mitochondrial phenotypes observed in $dCHCHD2^{H43}$ flies. We introduced LacZ, dCHCHD2 or hCHCHD2, including WT and two mutants,

into $dCHCHD2^{H43}$ flies; hCHCHD2 WT and mutant genes were expressed at similar levels (Fig. 6a). Reintroduction of dCHCHD2 and hCHCHD2 WT almost completely rescued the morphological defects of mitochondria in the indirect flight muscles,

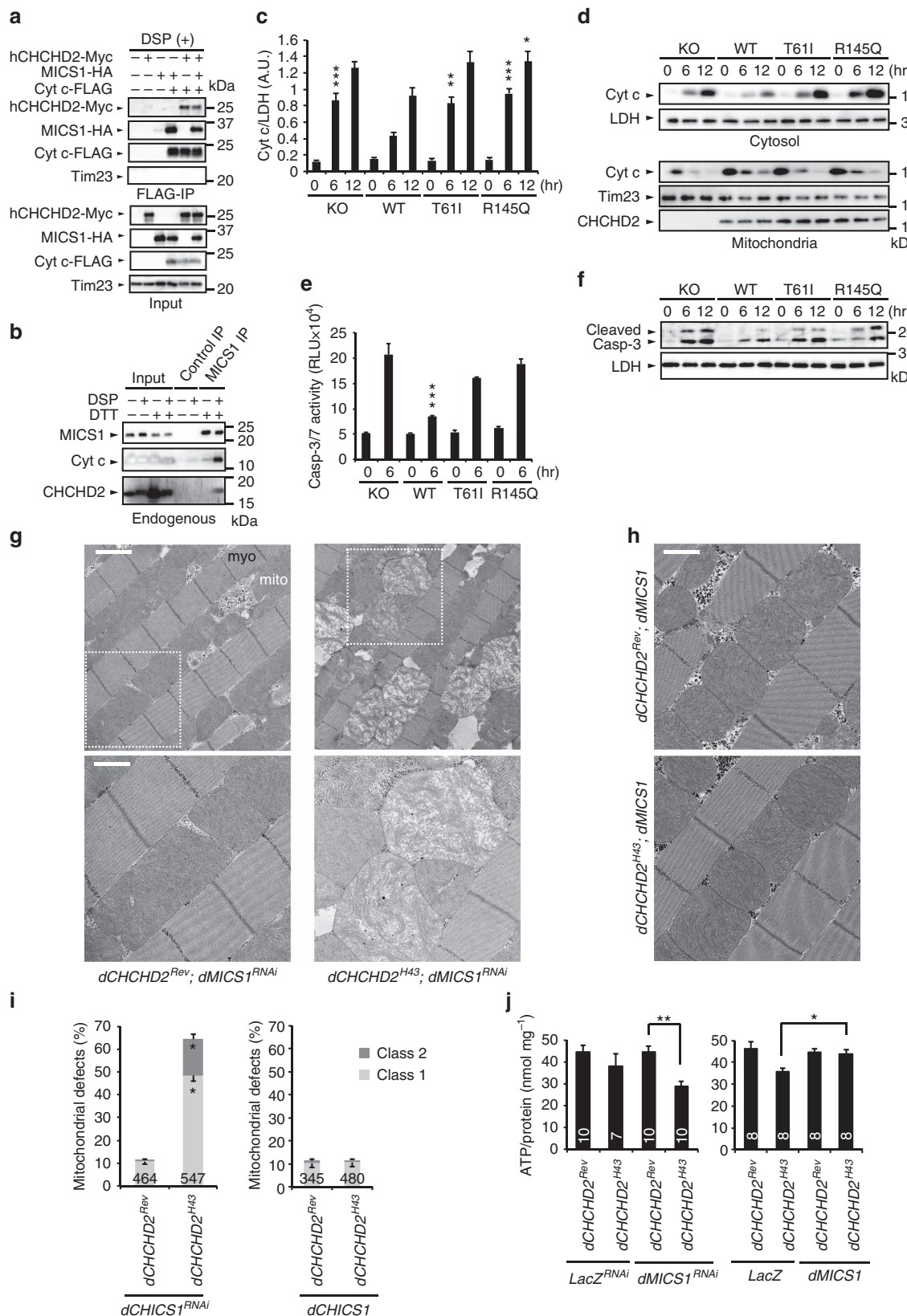

whereas the dilatation of matrix space, disordered cristae, and 'swirl' phenotype were still observed with a control LacZ, hCHCHD2 T61I or R145Q (Fig. 6b,c and Supplementary Fig. 8). Importantly, the age-dependent loss of PPL1 DA neurons observed in *dCHCHD2*[H43] flies was rescued by hCHCHD2 WT but not by the mutants T61I or R145Q (Fig. 6d). Consistent with the rescue of the mitochondrial morphology and the DA neuron loss, impairment of ATP production, locomotor defects, short lifespan, and sensitivity to oxidative stress observed in *dCHCHD2*[H43] flies were suppressed by dCHCHD2 or hCHCHD2 WT, whereas rescue effects were generally weaker in hCHCHD2 mutants, raising the possibility that these two missense mutations are loss-of-function mutations (Fig. 6e–g). To test this idea, hCHCHD2 T61I and R145Q were expressed in *dCHCHD2* heterozygous flies. *dCHCHD2* heterozygous flies expressing LacZ exhibited almost normal mitochondrial morphology and showed slightly decreasing ATP levels compared to a normal control (Fig. 7a–c). Overexpression of PD-associated mutants did not produce morphological defects, and ATP levels were comparable to the heterozygous mutant with LacZ (Fig. 7a–c). We found that the T61I mutation, which occurs in a central conserved region (Supplementary Fig. 1a), destabilized the CHCHD2 protein, leading to its appearance in the insoluble fraction when expressed in bacteria (Fig. 7d). CHCHD2 formed a homodimer (Fig. 7e). R145Q, which is located adjacent to the second $Cx_9C$ motif (Supplementary Fig. 1a), affected the dimer formation (Fig. 7e). These genetic and molecular analyses suggest that the two mutations affect the normal functions of CHCHD2 showing loss-of-function properties.

## 4E-BP rescues neurodegeneration by loss of CHCHD2.

Our data suggest that loss or mutation of CHCHD2 produces ROS due to the incomplete OXPHOS activity, leading to oxidative stress and subsequent neurodegeneration. Supporting this idea, a compensatory upregulation of 4E-BP, which has a protective role against oxidative stress maintaining mitochondrial integrity, was observed in *dCHCHD2*-deficient flies (Fig. 2g,h)[12,14,31]. Overexpressing CHCHD2 and LacZ with the GAL4-UAS system caused 4E-BP expression to increase slightly, probably due to GAL4, LacZ and/or CHCHD2 overexpression. However, reintroduction of dCHCHD2 at a level similar to endogenous levels suppressed 4E-BP induction in *dCHCHD2*-deficient flies (Fig. 8a and Supplementary Fig. 9a). Under the same expression conditions as the dCHCHD2 introduction, WT hCHCHD2 successfully suppressed 4E-BP induction in *dCHCHD2*-deficient flies, whereas PD mutants failed to do so, suggesting that mitochondrial stress was alleviated by intact CHCHD2 (Fig. 8a).

We then introduced 4E-BP into *dCHCHD2*[−/−] flies because 4E-BP has multiple beneficial roles in mitochondria. Ubiquitous expression of 4E-BP lessened the ATP decline in *dCHCHD2*[−/−] flies (Fig. 8b). A similar effect was obtained by treating with rapamycin, which activates 4E-BP through inhibition of a 4E-BP kinase target of rapamycin (TOR) (Fig. 8b)[17]. Consistent with the effect on ATP, mitochondrial morphology and protein oxidization were also improved by 4E-BP overexpression (Fig. 8c–e). Dopaminergic and serotoninergic expression of 4E-BP in *dCHCHD2*[H43] flies extended the lifespan and improved the motor activity and DA neuron loss to the extent of Rev expressing LacZ or 4E-BP (Fig. 8f–h).

Although DJ-1 and SOD1 have anti-oxidative activity, we could not detect an obvious genetic interaction between *CHCHD2* and *DJ-1* or between *CHCHD2* and *SOD1* in *Drosophila*[18,22]. Loss of DJ-1 did not exacerbate mitochondrial morphology in *dCHCHD2*-deficient flies (Supplementary Fig. 9b). DJ-1 or SOD1 overexpression did not extend the lifespan of *dCHCHD2*-deficient flies (Supplementary Fig. 9c,d). In addition, SOD1 failed to improve the climbing defect, TH neuron loss of *dCHCHD2*-deficient flies (Supplementary Fig. 9e,f), suggesting that simple redox control is not sufficient for mitochondrial protection. We also tested the genetic association of *CHCHD2* with *PINK1* and *parkin*, both of which are involved in mitochondrial quality control in the same signalling pathway[32]. Although dCHCHD2 is upregulated in *PINK1*- and *parkin*-deficient flies, there was no evidence that loss of CHCHD2 exacerbates the mitochondrial phenotypes caused by the loss of *PINK1* or *parkin* (Fig. 9a) or that CHCHD2 regulates mitochondrial size or dynamics, as observed in PINK1 and Parkin signalling (Fig. 9b). However, overexpression of PINK1 or Parkin led to much sharper reductions in ATP and stronger changes in mitochondrial morphology caused by the loss of dCHCHD2, whereas PINK1 or Parkin overexpression alone promoted the fragmentation of mitochondria with normal crista integrity (Fig. 9c,d).

## Discussion

Mitochondrial dysfunction is a prominent feature of PD. The newly identified PD-related gene *CHCHD2*, which encodes a mitochondrial protein, could provide new insight into the aetiology of PD. In this study, we characterized the physiological and pathological roles of CHCHD2 using *Drosophila* and mouse cultured cells. Although *CHCHD2*-deficient flies and mice develop normally, age-dependent mitochondrial defects, increased oxidative stress, and DA neuron loss were detected in *Drosophila*. Previous studies suggest that CHCHD2 has a role in OXPHOS function. During OXPHOS, electrons are extracted

**Figure 5 | CHCHD2 interacts with MICS1 and Cyt c.** (**a**) CHCHD2 interacts with MICS1 and Cyt c in HEK293T cells. Co-precipitated proteins with Cyt c-FLAG were detected after DSP treatment. An inner mitochondrial membrane protein, Tim23 served as a control. (**b**) Endogenous interaction of MICS1 with CHCHD2 and Cyt c. Lysates from cells treated with or without DSP were immunoprecipitated with anti-MICS1 or a control antibody. Co-precipitated proteins were analysed in the presence or absence of DTT. (**c**) CHCHD2 negatively regulates Cyt c release. An empty vector (KO), hCHCHD2 WT, T61I or R145Q was virally introduced into *CHCHD2*-deficient MEFs. Cells were treated with 500 nM actinomycin D for the indicated periods of time. The amounts of cytosolic Cyt c (mean ± s.e.m.) were determined by western blot and were normalized to lactate dehydrogenase (LDH) levels. *$P = 0.047$, **$P = 0.0013$, ***$P < 0.001$ versus the same period of WT from eight independent experiments (one-way ANOVA with Tukey–Kramer test). (**d**) A representative western blot of the Cyt c release is shown. LDH and Tim23 serve as cytosolic and mitochondrial loading controls, respectively. (**e,f**) CHCHD2 suppresses caspase activation by actinomycin D. (**e**) Caspase-3/7 activity (mean ± s.e.m.) was measured by a chemiluminescence-based assay kit. RLU, relative light unit. ***$P < 0.0001$ versus 6 h of all other groups. $n = 9$ from two independent experiments (one-way ANOVA with Tukey–Kramer test). (**f**) Cleaved caspase-3 was detected by western blot. (**g,h**) TEM images of the indirect flight muscles of 7-day-old adult flies with the indicated genotypes are shown. (upper) Mitochondrion, mito; myofibril, myo. (Lower) Higher-magnification images of boxed regions in upper panels. Scale bars, 2 μm (upper in **g**) and 1 μm (lower in **g** and **h**), respectively. (**i**) Mitochondrial defects were counted as shown in Fig. 1e (mean ± s.e.m.). $n = 345$–547 from three independent samples. *$P < 0.0001$ versus the same class of *dCHCHD2*[Rev] (two-tailed student's t-test). (**j**) ATP contents of 30-day-old flies were measured as in Fig. 1f. Mean ± s.e.m., $n = 7$–10. *$P = 0.046$, **$P < 0.001$ (one-way ANOVA with Dunnett's test). Transgenes were driven by the *MHC-GAL4* (**g–j**). ANOVA, analysis of variance; LDH, lactate dehydrogenase, TEM, transmission electron microscopy.

from substrates and are then transferred to molecular oxygen through a chain of enzymatic complexes, I–IV. In the final step, complex IV (Cyt c oxidase) ensures the reduction of molecular oxygen to water, without formation of oxygen radicals. CHCHD2 binds to Cyt c in conjunction with MICS. Cyt c, which transfers electrons from complex III (ubiquinol-Cyt c reductase) to complex IV, is localized in the mitochondrial intermembrane space. When apoptotic signalling is activated by proapoptotic Bcl-2 family members, such as Bax or tBid, Cyt c is released

through the permeabilized mitochondrial outer membrane. Knockdown of MICS1 and CHCHD2 facilitates Cyt c release upon treatment with apoptosis-inducing reagents and UV irradiation[7,30]. Considering our findings together with the above studies, we hypothesize that CHCHD2 and MICS1 stabilize Cyt c to maintain normal electron flow during OXPHOS (Fig. 10). When Cyt c function is compromised, electron flow from complex III to complex IV is impaired, and excess electrons from Cyt c may react with molecular oxygen,

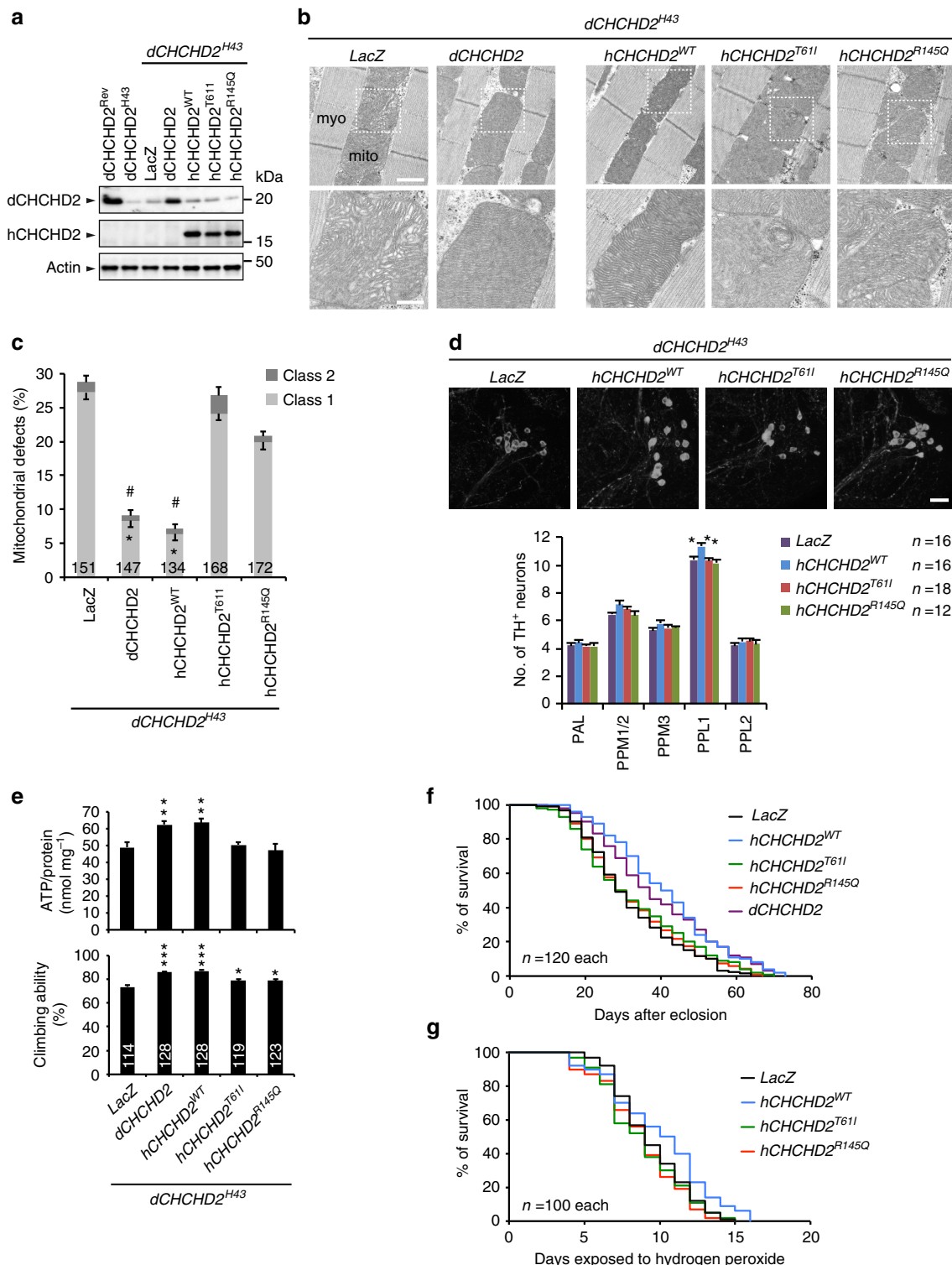

resulting in the production of superoxide and hydrogen peroxide, a major cause of oxidative stress. Supporting this idea, *dCHCHD2*-deficient flies exhibit various phenotypes associated with oxidative stress, including increased 4-HNE, 8-OHdG and protein/GSH oxidization and upregulation of 4E-BP. In these flies, aging is a key factor in mitochondrial degeneration and DA neuron loss. An age-related decline in the activity of complex IV, but not other respiratory complexes, has been reported without detectable alteration of complex IV protein levels in *Drosophila*[33]. This observation raises the possibility that the function of complex IV, in addition to that of Cyt c, is compromised in *dCHCHD2*-deficient flies as they aged, resulting in a high incidence of ROS.

Although multiple lines of evidence indicate that chronic oxidative stress occurs by CHCHD2 loss, overexpression of SOD1 and DJ-1 or removal of *DJ-1* did not affect *dCHCHD2* phenotypes, which strongly suggests that oxidative stress is only part of the pathogenesis of PD that is linked to *CHCHD2*. The functional difference between SOD1/DJ-1 and 4E-BP is that the latter regulates mitochondrial functions and proteostasis in addition to upregulation of anti-oxidative proteins such as anti-oxidant GST-S1 (refs 11,12,17,31). For mitochondrial regulation by 4E-BP, it has been demonstrated that 4E-BP is involved in translational upregulation of complexes I and IV subunits under dietary restriction, which enhances OXPHOS[12]. Another study has proposed that 4E-BP suppresses OXPHOS proteins and mitochondrial respiration[31]. We favour the latter idea because hypoxia improves the mitochondrial defects of *dCHCHD2* flies (Supplementary Fig. 10a,b).

In sharp contrast to 4E-BP overexpression, PINK1 or Parkin overexpression decreased the mitochondrial integrity of *dCHCHD2*-deficient flies. According to the hypothesis that PINK1-Parkin signalling is responsible for mitochondrial quality control, PINK1 or Parkin overexpression might activate mitophagy. This effect would cause most of the mitochondria in *dCHCHD2*-deficient flies to be recognized as damaged mitochondria, leading to lysosomal degradation. Nevertheless, given that CHCHD2 responds to a variety of mitochondrial stresses, including *PINK1*, *parkin* and *DJ-1* mutations, CHCHD2 could be a potential biomarker of PD associated with mitochondrial dysfunction.

Because the mitochondrial respiratory chains are localized at the cristae, crista structure is closely associated with OXPHOS activity. CHCHD10 has been reported to be enriched at the crista junctions, and expression of a CHCHD10 mutant, which causes frontotemporal dementia-amyotrophic lateral sclerosis, led to fragmentation of the mitochondrial network and defects in crista structures[4]. Although our electron microscopic study using *CHCHD2*-deficient MEFs did not detect obvious abnormalities in crista organization due to a variety of mitochondrial morphologies in the cytosol (Supplementary Fig. 11), the lack of *dCHCHD2* led to moderate crista deformation in the indirect flight muscles of *Drosophila*. Because MICS1 has also been reported to regulate crista organization[30], CHCHD2 along with MICS1 might maintain crista integrity to accommodate Cyt c properly.

PD caused by missense mutations in *CHCHD2* is inherited as a dominant trait[1]. This suggests that neurodegeneration may be caused by gain-of-function mutations. Our study indicates that human pathogenic T61I and possible R145Q mutants do not fully rescue the mitochondrial defects caused by the lack of dCHCHD2, suggesting that these mutations compromise its physiological functions. Supporting this idea, we found that CHCHD2 harbouring PD mutations exhibits reduced solubility or impaired homodimerization, although it should be determined whether dimerization is important for its physiological function in future research. Importantly, the expression of T61I and R145Q in *dCHCHD2* heterozygous flies did not produce mitochondrial defects, which strongly suggests that the aetiology of PD linked to *CHCHD2* includes a loss-of-function aspect. However, further studies should determine whether gain-of-function events are also involved in neurodegeneration by CHCHD2 mutations.

CHCHD2 levels were altered by various mitochondrial stresses. A previous study showed that CHCHD2 transactivates *COX4I2* and *CHCHD2* itself in the nucleus[3,34]. Thus, the upregulation of CHCHD2 due to mitochondrial stress might contribute to the transcriptional regulation of these genes, resulting in a more complex pathogenesis than we hypothesize here. Hyperoxia suppressed dCHCHD2 levels in *Drosophila*, which suggests that hypoxia-inducible factor (HIF) is involved in its regulation. However, we could not find the consensus sequence of the hypoxia-responsive element upstream of the *dCHCHD2* gene. In addition, hypoxia or knockdown of HIF-1α/Sima or HIF-1ß/Tango did not change dCHCHD2 levels (Supplementary Fig. 12). Cyclic adenosine monophosphate response element and a novel oxygen-responsive element have been proposed as regulation sequences for *CHCHD2* gene expression in mammals[3,35]. Although we could not find putative sequences for these elements in the 5′-untranslated region of the *dCHCHD2* gene, further analysis is needed to clarify the involvement of these elements in *Drosophila*.

In summary, we propose that CHCHD2 controls OXPHOS through regulation of Cyt c and crista integrity. Loss or mutation of CHCHD2 leads to oxidative stress probably through dysregulation of electron flow in OXPHOS. Our study sheds light on the aetiology of PD linked to *CHCHD2* and potential therapeutic targets in PD caused by mitochondrial dysfunction.

**Figure 6 | PD-associated mutations of hCHCHD2 affect mitochondrial functions and DA neuron survival. (a)** Expression of hCHCHD in *dCHCHD2^H43* flies. LacZ, dCHCHD2, and hCHCHD2 (WT, T61I and R145Q), were expressed in *dCHCHD2^H43* flies, and thorax muscle tissues were subjected to western blot. Actin, a loading control. **(b)** TEM images of the indirect flight muscles of 14-day-old flies with the indicated genotypes. (Lower) Higher-magnification images of boxed regions in upper panels. Scale bars, 2 μm (upper) and 500 nm (lower), respectively. **(c)** Mitochondria with abnormal cristae or showing degeneration were quantified as shown in Fig. 1e (mean ± s.e.m.). $n = 134$–172 from three independent samples. *$P < 0.0001$ versus class 1 of LacZ; # $P < 0.0001$ versus total defects (class 1 + class 2) of LacZ (one-way ANOVA with Tukey–Kramer test). **(d)** Expression of hCHCHD2 WT but not PD mutants rescues the loss of PPL1 DA neurons in *dCHCHD2^H43* flies. Fifty-day-old flies were analysed (mean ± s.e.m., $n = 12$–18), and representative images of the PPL1 DA neurons visualized with anti-dTH are shown. Scale bar, 10 μm. *$P < 0.026$ versus PPL1 of hCHCHD2 WT (one-way ANOVA with Tukey–Kramer test). **(e)** ATP reduction and climbing defects in *dCHCHD2^H43* flies were rescued by dCHCHD2 and hCHCHD2 WT, whereas hCHCHD2 T61I and R145Q did not fully suppress these phenotypes. Thirty-day-old flies were analysed (mean ± s.e.m.). $n = 15$ for ATP assay; 20 trials with 114–128 flies from five independent experiments for the climbing assay. *$P < 0.05$, **$P < 0.01$, ***$P < 0.001$ versus LacZ (one-way ANOVA with Tukey–Kramer test). **(f)** dCHCHD2 ($P < 0.001$ by log-rank test) and hCHCHD2 WT ($P < 1^{-11}$ by log-rank test), but not PD mutants, extend the lifespan of *dCHCHD2^H43* flies. No differences with LacZ, T61I, and R145Q. **(g)** hCHCHD2 WT ($P < 0.05$ by log-rank test), but not PD mutants, improved the sensitivity to oxidative stress by peroxide. No differences with LacZ, T61I and R145Q. All transgenes were driven by *Da-GAL4,* and *LacZ* served as a negative control. ANOVA, analysis of variance; TEM, transmission electron microscopy.

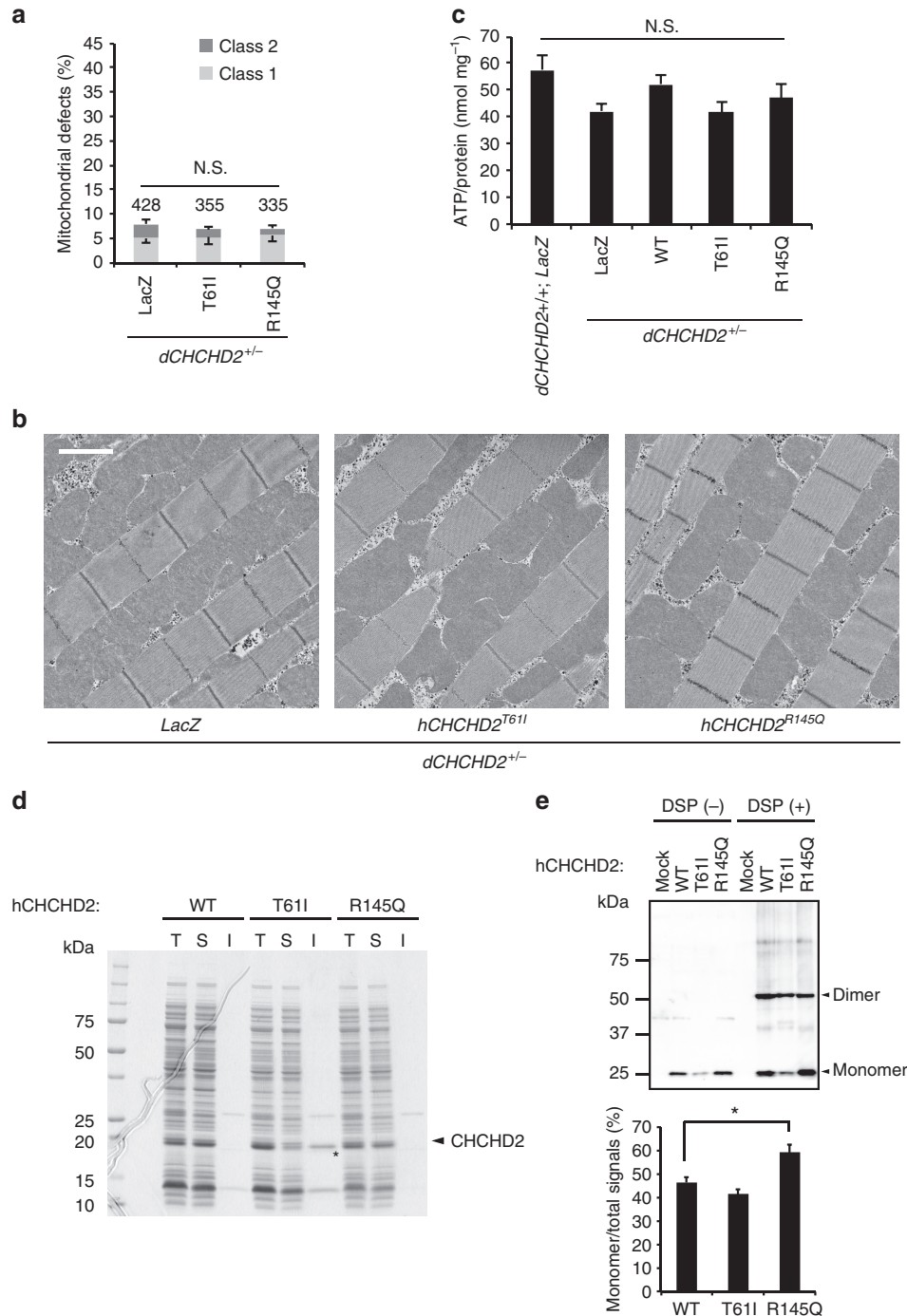

**Figure 7 | Biochemical properties of hCHCHD2 are altered in PD mutants.** (**a**) Mitochondrial defects in 14-day-old adult flies, defined as classes 1 and 2 defects, were counted as shown in Fig. 1e (mean ± s.e.m.). Transgenes (LacZ, T61I and R145Q) were driven in *dCHCHD2* heterozygous flies by *Da-GAL4*. $n = 335$–428 from three independent samples. N.S., not significant by one-way ANOVA with Tukey–Kramer test. (**b**) Representative TEM images of the mitochondria in the indirect flight muscles counted in **a**. Scale bar, 2 μm. (**c**) ATP content in the thorax muscle tissues of 14-day-old adult flies with the indicated genotypes. Mean ± s.e.m. from seven independent samples. N.S., not significant by one-way ANOVA with Tukey–Kramer test. (**d**) Bacterially expressed hCHCHD2 T61I has difficulty in solubilizing. hCHCHD2 WT and PD mutants were expressed in SHuffle T7 Express *Escherichia coli* (New England Biolabs), which promotes disulfide bond formation in the cytoplasm. Total lysate (T), phosphate-buffered saline-soluble (S) and insoluble (I) protein fractions solubilized in SDS sample buffer containing 2% SDS were separated on SDS–PAGE and were visualized by Coomassie brilliant blue staining. An asterisk indicates insoluble hCHCHD2 T61I, whereas there were no insoluble hCHCHD2 bands in WT and R145Q. (**e**) hCHCHD2 forms a dimer. hCHCHD2-expressing MEFs treated with or without 0.5 μg ml$^{-1}$ DSP in PBS for 20 min at RT were lysed and subjected to western blot analysis with anti-CHCHD2. The ratio of monomer signals to total signals was graphed (mean ± s.e.m., $n = 6$ in each group). A monomer form of hCHCHD2 WT and T61I accounts for less than 50% of total CHCHD2 signals under this experimental condition, whereas the R145Q monomer form was higher (~60%, *$P = 0.027$ by one-way ANOVA with Tukey–Kramer test). ANOVA, analysis of variance; TEM, transmission electron microscopy.

## Methods

**Antibodies.** Recombinant 6xHis-GST-tagged dCHCHD2 was purified from bacteria and used for antibody production (Evebio Science, Japan). Antibodies used in western blot analysis were as follows: anti-dCHCHD2 (1:1,000 dilution; in-house), anti-hCHCHD2 (1:1,000; Proteintech, 19424-1-AP), anti-FLAG

(1:1,000; Sigma-Aldrich, clone M2), anti-Myc (1:1,000; Millipore, clone 4A6), anti-HA (1:1,000; Roche, clone 3F10), anti-d4E-BP (1:1,000; in-house[16]), anti-MICS1 (1:1,000; Everest Biotech, EB09815), anti-cleaved caspase-3 (1:1,000; Cell Signaling Technology, #9664), anti-Cyt c (1:1,000; Abcam, 37BA11), anti-LDH (1:1,000; Santa Cruz, H-160), anti-Tim23 (1:2,000; BD, clone 32/Tim23),

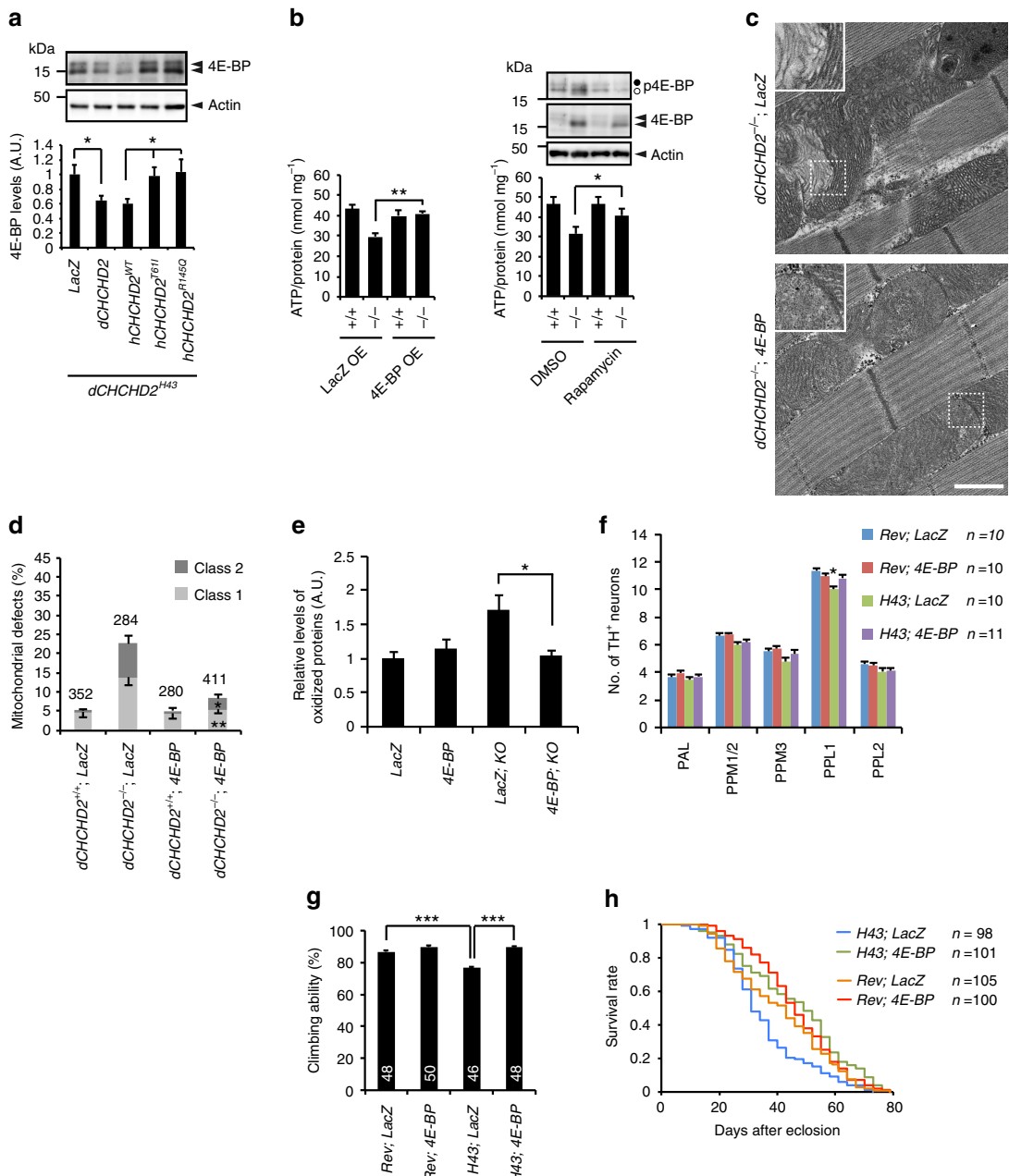

**Figure 8 | 4E-BP rescues neurodegeneration by loss of CHCHD2.** (**a**) hCHCHD2 WT, but not PD mutants, suppresses 4E-BP expression. dCHCHD2 and hCHCHD2 were induced in adult flies with 2.0 µg ml[−1] RU486 for 3 days (Supplementary Fig. 9a). *UAS-LacZ* crossed with *Da-GS* served as a negative control. 4E-BP levels (mean ± s.e.m., $n = 6$) were quantified and normalized to actin levels. *$P = 0.038$; LacZ versus dCHCHD2, *$P = 0.029$; hCHCHD2 WT versus T61I, *$P = 0.013$; hCHCHD2 WT versus R145Q. (**b–h**) 4E-BP rescues *dCHCHD2* phenotypes. (**b**) ATP content in thorax muscle tissues from 14-day-old flies. 4E-BP or LacZ was ubiquitously expressed by using the *Da-GAL4* driver. Rapamycin treatment (50 µM) for 14 days, but not DMSO treatment, reduced the mildly (a white dot) and highly (a black dot) phosphorylated 4E-BP levels in western blots. Mean ± s.e.m. from seven independent samples. *$P = 0.014$, **$P = 0.003$. (**c**) TEM images of the indirect flight muscle of 40-day-old adult flies with the indicated genotypes and higher-magnification images of the boxed regions are shown. Scale bar = 1 µm. (**d**) Mitochondria with abnormal cristae or degenerating mitochondria in **c** were counted as shown in Fig. 1e (mean ± s.e.m.). $n = 280–411$ from three independent samples. *$P = 0.0021$, **$P < 0.0001$ versus *dCHCHD2*[−/−]; LacZ. (**e**) Detection of protein oxidation. Thoraxes of 45-day-old flies were used for the Oxyblot assay. Protein oxidation of *dCHCHD2*-deficient flies was suppressed by 4E-BP overexpression ($n = 5$, *$P = 0.02$). (**f–h**) 4E-BP or LacZ was expressed using the *Ddc-GAL4* driver in *dCHCHD2*[H43] and revertant flies, and flies were analysed for the number of remaining DA neurons (**f**, *$P = 0.047$ versus PPL1 of *H43; 4E-BP*, $n = 10–11$), climbing ability (**g**, ***$P < 0.001$; $n = 46–50$) and lifespan (**h**, $P < 0.04$, *H43; LacZ* versus any other groups by log-rank test). (**a,b,d–g**) Data represent the mean ± s.e.m. and were analysed by one-way ANOVA with Tukey–Kramer test. ANOVA, analysis of variance; TEM, transmission electron microscopy.

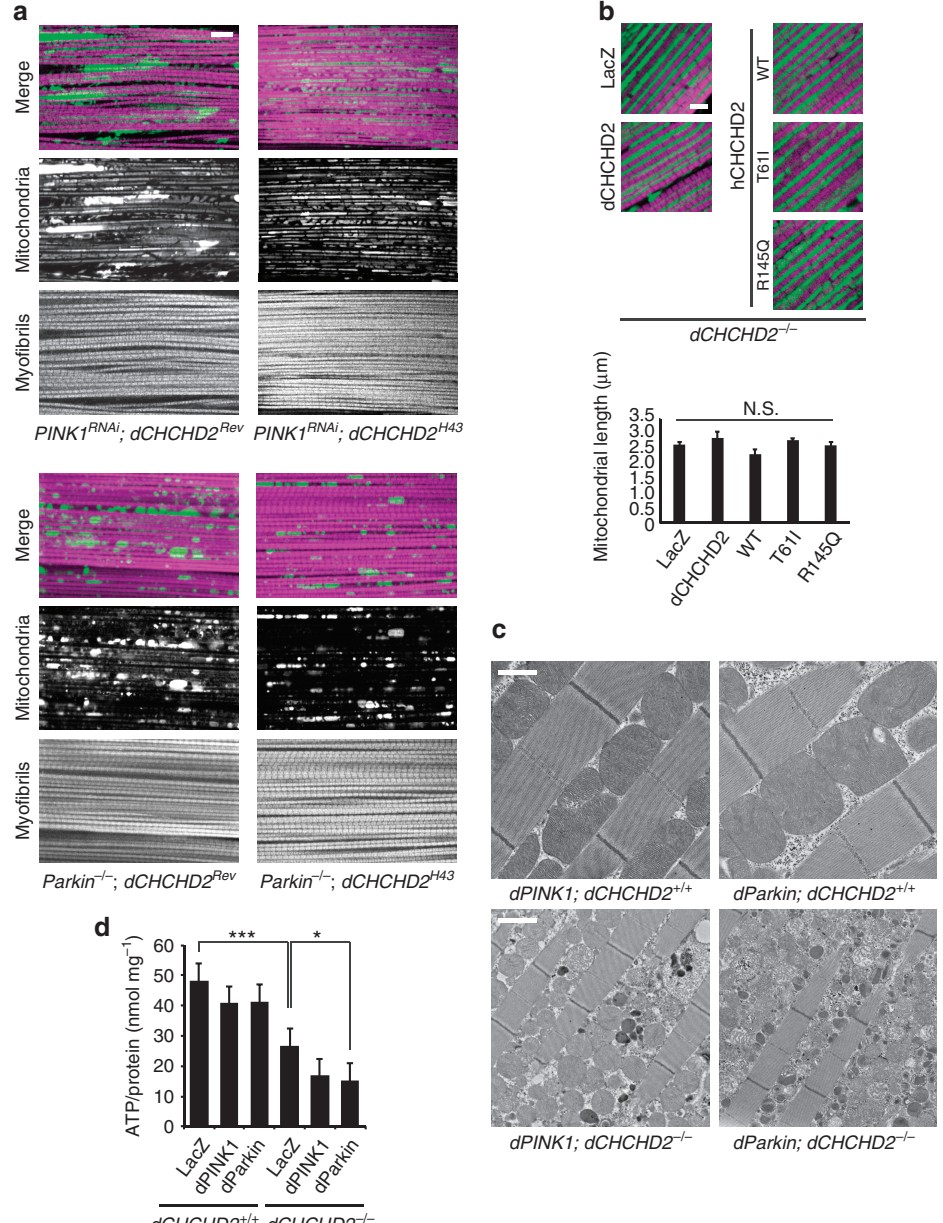

**Figure 9 | Effects of PINK1-Parkin signalling on the *dCHCHD2* mitochondrial phenotypes.** (**a**) Mitochondrial morphology of *PINK1* RNAi (upper) and *parkin*-deficient (lower) flies with or without dCHCHD2 activity. Fluorescent images of the indirect flight muscles of 7-day-old adult male flies with the indicated genotypes are shown. To visualize the mitochondria, the mitoGFP (green) transgene was co-expressed, and the muscle tissue was counterstained with phalloidin (magenta). Scale bar, 10 μm. (**b**) Loss of CHCHD2 and expression of PD mutants do not affect mitochondrial size. Fluorescent images of the indirect flight muscles of 14-day-old adult male flies with the indicated genotypes are shown, as in (**a**). Scale bar, 5 μm. A graph shows the length of the long axis of the muscle mitochondria (mean ± s.e.m.). $n = 9$ from three independent flies in each group. N.S., not significant ($P = 0.0629$) versus the LacZ group by Dunnett's test. (**c**) TEM images of the indirect flight muscle in the indicated genotypes of 14-day-old adult flies. Scale bars, 1 μm (upper) and 2 μm (lower), respectively. (**d**) ATP levels in the thorax muscle tissues in the indicated genotypes. $n = 8$. *$P = 0.045$, ***$P < 0.0001$ (one-way ANOVA with Tukey–Kramer test). The genotypes used were as follows: (**a**) *dCHCHD2^Rev^/Y; UAS-mitoGFP/+; MHC-GAL4, UAS-PINK1 RNAi/+* (*PINK1^RNAi^; dCHCHD2^Rev^*), *dCHCHD2^H43^/Y; UAS-mitoGFP/+; MHC-GAL4, UAS-PINK1 RNAi/+* (*PINK1^RNAi^; dCHCHD2^H43^*). *dCHCHD2^Rev^/Y; UAS-mitoGFP/+; Da-GAL4, Parkin^Δ21^/Parkin^1^* (*Parkin^−/−^; dCHCHD2^Rev^*), *dCHCHD2^H43^/Y; UAS-mitoGFP/+; Da-GAL4, Parkin^Δ21^/Parkin^1^* (*Parkin^−/−^; dCHCHD2^H43^*). (**b**) *dCHCHD2^null^/Y; UAS-mitoGFP/+; MHC-GAL4/UAS-LacZ* (*LacZ*), *dCHCHD2^null^/Y; UAS-mitoGFP/+; MHC-GAL4/UAS-dCHCHD2* (*dCHCHD2*), *dCHCHD2^null^/Y; UAS-mitoGFP/+; MHC-GAL4/UAS-hCHCHD2 WT* (*WT*), *dCHCHD2^null^/Y; UAS-mitoGFP/+; MHC-GAL4/UAS-hCHCHD2 T61I* (*T61I*), *dCHCHD2^null^/Y; UAS-mitoGFP/+; MHC-GAL4/UAS-hCHCHD2 R145Q* (*R145Q*). (**c,d**) *+/Y; MHC-GAL4/UAS-LacZ* (*LacZ; dCHCHD2^+/+^*), *+/Y; UAS-dPINK1; MHC-GAL4/UAS-dPINK1* (*dPINK1; dCHCHD2^+/+^*), *+/Y; UAS-dParkin/+; MHC-GAL4/+* (*dParkin; dCHCHD2^+/+^*), *dCHCHD2^null^/Y; MHC-GAL4/UAS-LacZ* (*LacZ; dCHCHD2^−/−^*), *dCHCHD2^null^/Y; UAS-dPINK1; MHC-GAL4/UAS-dPINK1* (*dPINK1; dCHCHD2^−/−^*), *UAS-dParkin/+; MHC-GAL4/+* (*dParkin; dCHCHD2^−/−^*). ANOVA, analysis of variance; TEM, transmission electron microscopy.

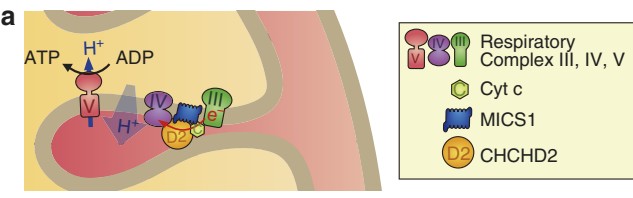

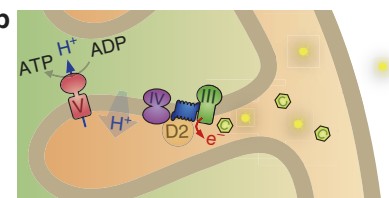

**Figure 10 | A model for roles of CHCHD2 in mitochondria. (a)** CHCHD2 in conjunction with MICS1 stabilizes Cyt c in the respiratory chains, allowing for proper electron flow from complex III to complex IV and consequent ATP production. (**b**) Mutation or absence of CHCHD2 destabilizes Cyt c in the respiratory chain, which causes electron leakage and ROS generation.

anti-Tom20 (1:500; Santa Cruz, FL-145), anti-complex I (1:10,000; Abcam, 17D95), anti-complex IV (1:1,000; Abcam, 20E8C12), anti-complex V (1:10,000; Abcam, 15H4C4), anti-SOD1 (1:1,000; Abcam, ab13498), anti-OXPHOS cocktail (1:250; Abcam, ab110413), anti-DJ-1b (1:1,000; in-house[22]), anti-actin (1:10,000; Millipore, MAb1501), and anti-tubulin (1:1,000; Sigma-Aldrich, clone DM1A). The following antibodies were used for immunocyto/histochemistry: anti-4-HNE (1:750; JaICA, HNEJ-2), anti-8-OHdG (1:20; JaICA, N45.1), anti-hCHCHD2 (1:200; Proteintech, 19424-1-AP), anti-FLAG (1:1,000; Sigma-Aldrich, clone M2), anti-Myc (1:1,000; Millipore, clone 4A6), anti-Myc (1:200; Cell Signaling Technology, #2278), anti-HA (1:1,000; Roche, clone 3F10), anti-Hsp60 (1:500; BD, clone 24/HSP60) and anti-dTH (1:250; in-house[16]).

***Drosophila* genetics.** Fly culture and crosses were performed on standard fly food containing yeast, cornmeal, and molasses, and the flies were raised at 25 °C. The $w^{1118}$ ($w^-$) or revertant line was used as a WT genetic background. Complementary DNA (cDNA) for dCHCHD2 or hCHCHD2 was subcloned into the *pUAST* vector, and transgenic lines were generated on the $w^-$ background (BestGene). For gene targeting of dCHCHD2, we used conventional mutagenesis by transposon mobilization of p{EPgy2}EY05234 (Bloomington stock 15790) and generated a hypomorphic $CHCHD2^{H43}$ allele and its revertant allele as a WT control. We also employed CRISPR/Cas9 technology to generate a *dCHCHD2* null allele using tandem guide RNA plasmids (pCFD4) and a fly line expressing Cas9 during oogenesis (Bloomington stock 54591)[36]. Two protospacers corresponding to sequences of dCHCHD2 were cloned into pCFD4 digested with BbsI using an In-Fusion HD cloning kit (Takara Bio) with the following primer pairs: 5′-TATATAGGAAAGATATCCGGGTGAACTTCGGGGACTGGCTGAAC-GTCCAGTTTTAGAGCTAGAAATAGCAAG-3′, 5′-ATTTTAACTTGCTATTT-CTAGCTCTAAAACTCTACTGGAGATGGTGGCTCGACGTTAAATTGAAAA-TAGGTC-3′. To avoid potential off-target mutations generated by Cas9, the *dCHCHD2*-deficient line was backcrossed to the $w^-$ background for six generations, and $w^-$ was used as a WT allele.

All other fly stocks and GAL4 lines used in this study were obtained from the Bloomington *Drosophila* Stock Center, Vienna *Drosophila* Resource Center, Kyoto Stock Center, NIG-fly (National Institute of Genetics, Japan), FlyORF and have been previously described: UAS-PolG$_{III}$ (ref. 24); UAS-OTC and UAS-ΔOTC[25]; UAS-d4E-BP[16]; and daughterless–Gene-Switch[37]. The *Drosophila* stocks used in the experiments in the Supplementary Information were as follows: DJ-1b Δ93 (ref. 18); UAS-dPINK1 and UAS-PINK1$^{RNAi}$ (ref. 38); Parkin$^{Δ21}$ (ref. 39); Parkin$^1$ (ref. 40); UAS-dParkin[41]; UAS-DJ-1b and UAS-human DJ-1 (ref. 22).

**Whole-mount immunostaining and electron microscopy analysis.** The mitochondrial morphology of the indirect flight muscles and the number of TH-positive neurons were analysed by whole-mount immunostaining as previously described[20]. Transmission electron microscopy images were obtained at the Laboratory of Ultrastructural Research of Juntendo University and Tokai Electron Microscopy, Inc. Abnormal mitochondria were counted and graded in a blind manner by C.Y. and H.M.

**TUNEL assay and muscle histochemistry.** The thorax samples embedded in paraffin blocks were sectioned from the lateral side at a thickness of 3 μm, and the serial sections were used for TUNEL assays, which made use of an *in situ* cell death detection kit (Sigma-Aldrich). Sections were also used for toluidine blue staining.

**Focused ion beam scanning electron microscopy tomography.** The indirect flight muscles of 14-day-old flies were subjected to focused ion beam scanning electron microscopy analysis using a Helios 660 dual-beam microscope (FEI). Epon blocks of tissue samples were repeatedly milled through focused ion beam in 20 nm steps, and the block face was imaged via scanning electron microscopy at an acceleration voltage of 3.0 kV using back-scatter electron signals. The acquired images were processed using Photoshop (Adobe Systems) to adjust the contrast and were converted to movies at 5 frames per second using ImageJ.

**ATP measurements.** The ATP content in the thorax muscles of *Drosophila* was measured as previously described[41]. Briefly, the thoraxes homogenized in 20 μl of homogenization buffer (6 M guanidine-HCl, 100 mM Tris and 4 mM EDTA (pH 7.8)) were centrifuged at 16,000 g. The supernatants diluted 1:1,000 and 1:10 with water were subjected to measurements of ATP concentrations and protein concentrations, respectively. ATP and protein concentrations were determined using the CellTiter-Glo luminescent cell viability assay kit (Promega) and the BCA protein assay kit (Pierce), respectively.

**Western blot analysis.** Fly heads and thoraxes homogenized in 40 μl of 3× SDS sample buffer using a motor-driven pestle were centrifuged at 16,000 g for 10 min and the supernatants were subjected to western blotting. For gene induction by Gene-Switch system, newly eclosing flies carrying the Da-GS driver were cultured with normal fly food containing various concentrations of RU486 for 3 days, and the gene induction was analysed in the thoraxes by western blot analysis as previously described[41]. Uncropped images of the western blotting are displayed in Supplementary Figs 13–15.

**OxyBlot assay and GSH/GSSG measurement.** For detection of protein oxidation, an OxyBlot Protein Oxidation Detection Kit (Merck Millipore) was used according to the manufacturer's instructions. The band intensity was analysed using ImageJ software. GSH/GSSG ratios from twenty male flies were measured by using a glutathione assay kit (Cayman Chemical), following a protocol provided by the manufacturer.

**Plasmids and cell culture.** Human CHCHD2 cDNAs were inserted in the pcDNA3 vector containing FLAG or Myc-His tag and in the pMXs-puro vector, followed by sequencing of the entire gene. Expression vectors for MICS1-HA and Cyt c-FLAG were a kind gift from Dr T. Oka. MEFs were retrovirally transfected with pMXs-puro harbouring hCHCHD2, and the transfected cells were then selected with 1 μg ml$^{-1}$ puromycin. The source of HEK293T cells was described elsewhere[43] and HeLa cells were purchased from RIKEN cell bank. They are maintained at 37 °C with a 5% CO$_2$ atmosphere in Dulbecco's modified Eagle's medium (DMEM, Sigma-Aldrich) supplemented with 10% FCS (Gibco), GlutaMax (Gibco), non-essential amino acids (Gibco) and 1% penicillin-streptomycin. The plasmids were transfected using Lipofectamine 2000 reagent (ThermoFisher Scientific) according to the manufacturer's instructions. The total amount of transfected cDNA was adjusted with vector DNA in every transfection experiment. All the cells used here were tested and negative for mycoplasma contamination.

**Identification of CHCHD2-binding proteins and binding assay.** For identification of CHCHD2-binding proteins, HEK293T cells expressing C-terminally FLAG-tagged human CHCHD2 were homogenized in cell lysis buffer (20 mM HEPES pH 7.5, 150 mM NaCl, 50 mM NaF, 1 mM Na$_3$VO$_4$, 0.5% digitonin, 1 mM PMSF, 5 μg ml$^{-1}$ leupeptin, 5 μg ml$^{-1}$ aprotinin and 3 μg ml$^{-1}$ pepstatin A). CHCHD2-FLAG was purified from the cell lysate with anti-FLAG M2 affinity gel (Sigma-Aldrich) and eluted with FLAG peptide (Sigma-Aldrich). The elution fraction was subjected to trichloroacetic acid precipitation and subsequent digestion with lysyl endopeptidase (Lys-C, Wako Chemicals USA). Samples were analysed on a liquid chromatography system coupled to a time-of-flight mass spectrometer (Q-STAR XL, AB Sciex)[44].

For co-immunoprecipitation assay using chemical crosslinker, cells were harvested and washed twice with PBS and then incubated with 0.5 mg ml$^{-1}$ DSP (ThermoFisher Scientific) at 25 °C for 15–20 min. After washing with TBS, the cells were solubilized at 4 °C for 30 min in cell lysis buffer (50 mM Tris-HCl pH 8.0, 150 mM NaCl, 1% Triton X-100 and 0.1% SDS) and then centrifuged to remove insoluble precipitates. The resultant supernatant was incubated with anti-FLAG or anti-Myc beads (Medical & Biological Laboratories) for 3 h at 4 °C, washed twice with PBS containing 1% Triton X-100, and then washed once with PBS alone. The precipitates were subjected to SDS–polyacrylamide gel electrophoresis (SDS–PAGE) and subsequent western blot analysis.

**Preparation of MEFs from *CHCHD2*-deficient mice.** CHCHD2$^{-/-}$ mice were generated by Unitech in Japan. In brief, the targeting vector was constructed to replace exons 2 and 3 with the neomycin resistance gene. The targeting vector was introduced into C57BL/6 embryonic stem (ES) cells by electroporation. After selection with G418, ES clones were subject to PCR screening and Southern blotting to identify homologous recombinant clones. Recombinant ES cells were

microinjected into C57BL/6 blastocysts to obtain chimeric mice, which were then mated to generate heterozygous and homozygous mice. Animal protocols were performed in accordance with the Juntendo University Animal Experiment Committee. $CHCHD2^{-/-}$ MEFs were prepared from embryos at 13.5–15.5 d.p.c. and were cultured at 37 °C with a 5% $CO_2$ atmosphere in DMEM containing with GlutaMax, supplemented with 10% fetal bovine serum, gentamicin (50 µg ml$^{-1}$; Gibco), and 1% non-essential amino acids.

**Cyt c release and caspase assay.** MEFs treated with 500 nM actinomycin D or 50 nM peroxide were suspended in mitochondrial isolation buffer (220 mM mannitol, 70 mM sucrose, 20 mM HEPES-KOH pH 7.4, 1 mM EDTA and protease inhibitor cocktail (Roche)), then homogenized by 20 passages through a 26-G syringe (Terumo) on ice[45]. Homogenates were centrifuged at 700 g for 10 min at 4 °C to obtain a post-nuclear supernatant. The post-nuclear supernatant was further centrifuged at 12,000 g for 15 min. Pellets were washed several times with the mitochondrial isolation buffer. Caspase activity was estimated with a Caspase-Glo 3/7 assay kit (Promega) or western blot with anti-cleaved caspase-3.

**Coupling assay of mitochondria in Drosophila embryonic cells.** Drosophila embryos (~800 in each) at the blastoderm stage collected 2–3 h post laying at 25 °C were dechorionated with 40% bleach for 2 min. Embryos were then homogenized in Schneider's insect medium containing 5% FCS using a Dounce tissue homogenizer by 10–12 up and down strokes with a loose pestle. Embryonic cells filtered with a 40-µm cell strainer were plated onto 24-well XF cell culture microplates (Seahorse Bioscience) pre-coated with Cell-Tak (Corning) at 80,000 cells per well and were subjected to a MitoStress test using the XF24 analyser (Seahorse Bioscience) at 29 °C.

**MitoSOX analysis in Drosophila embryonic cells.** Drosophila embryonic cells prepared as described above were stained with 20 µM MitoSOX (ThermoFisher Scientific) and 1 µg ml$^{-1}$ Hoechst 33342 for 20 min at RT. Fluorescence intensities of MitoSOX in cells ($1 \times 10^4$) were measured using LSRFortessa (BD Biosciences), and Hoechst 33342-positive cells were analysed using FlowJo version 9 (TreeStar).

**Mitochondrial respiratory chain complex activity assay.** Mitochondria from the thoracic muscles of 14-day-old adult flies were isolated by a reported protocol with some modifications[33]. Briefly, isolated thoraces from 20 flies were gently crushed in 200 µl of ice-cold isolation buffer (320 mM sucrose, 10 mM EDTA and 10 mM Tris-HCl pH 7.3) using a plastic pestle homogenizer and then spun twice at 500 g for 5 min at 4 °C to remove debris. After centrifugation for 10 min at 2,200 g, the pellet was resuspended in 100 µl of isolation buffer. The activities of complexes I–IV were measured following a previously reported protocol, which was adapted to the 96-well plate scale[46,47]. Briefly, for the measurement of the complex I activity, NADH dehydrogenase assay buffer (50 mM potassium phosphate buffer pH 7.5, 2.5 mg ml$^{-1}$ bovine serum albumin [BSA], 240 µM KCN, 70 µM decylubiquinone [dUb], 25 µM antimycin A) containing 2 µM rotenone or ethanol was divided into 100 µl aliquots in a microplate and left at RT for 5 min. Complex I activity was monitored by adding 0.4 µg of mitochondrial preparations and 200 µM NADH (2 mM NADH stock) to the reaction mix and following a decrease in absorbance at 340 nm (with reference wavelength at 425 nm). For the measurement of the complex II activity, succinate dehydrogenase assay buffer (25 mM potassium phosphate buffer pH 7.5, 2 mg ml$^{-1}$ BSA, 20 mM potassium succinate, 60 µM 2,6-dichlorophenolindophenol, 25 µM antimycin A, 2 mM KCN, 2 µM rotenone) with or without 10 mM malonic acid was divided into 100 µl aliquots in a microplate and left at RT for 5 min. Complex II activity was monitored by adding 0.4 µg of mitochondrial samples and 100 µM dUb to the reaction mix and following a decrease in absorbance at 600 nm. For the measurement of the complex III activity, cytochrome c reductase assay buffer (50 mM potassium phosphate buffer pH 7.5, 1 mM n-dodecyl β-D-maltoside [DDM], 1 mM KCN, 2 µM rotenone, 250 µM EDTA, and 1 mg ml$^{-1}$ BSA, 50 µM oxidized cytochrome c) with or without 25 µM antimycin A was divided into 100 µl aliquots in a microplate and left at RT for 5 min. The cytochrome c reductase activity was monitored by adding 200 µM decylubiquinol, and 0.075 µg of mitochondrial samples to the reaction mix and following an increase in absorbance at 550 nm. For the measurement of the complex IV activity, cytochrome c oxidase buffer (25 mM potassium phosphate pH7.0, 0.45 mM DDM and 100 µM reduced cytochrome c) with or without 2 mM KCN was divided into 100 µl aliquots in a microplate. The cytochrome c oxidase activity was monitored by adding 0.1 µg of mitochondrial samples to the reaction mix and following a decrease in absorbance at 550 nm. Each activity was evaluated by subtracting the activity in the presence of specific complex inhibitors, including rotenone, malonic acid, antimycin A and KCN.

**Binding assay of CHCHD2 to complex IV.** Expi293F cells (Thermo Fisher Scientific) were transfected with an expression plasmid for full-length human CHCHD2 c-terminally tagged with a PA tag[48] according to the manufacturer's protocol. The cells from a 50 ml culture were lysed via sonication in lysis buffer containing 50 mM Tris-HCl pH 7.5, 300 mM NaCl, 10% glycerol, 0.1% Triton-X

and cOmplete Ultra protease inhibitor cocktail (Roche), followed by centrifugation at 27,216 g for 15 min. The supernatant (10 ml in total) was incubated with 100 µl of anti-PA tag NZ-1-Sepharose for 1 h at 4 °C, and the beads were sequentially washed with 1 ml of wash buffer containing 50 mM Tris-HCl pH 7.5, 300 mM NaCl, 10% glycerol, followed by 1 ml of interaction buffer containing 50 mM Tris-HCl pH 7.5, 300 mM NaCl, 10% glycerol and 0.2% n-decyl-β-D-maltopyranoside (DM) (Anatrace, D322). The CHCHD2-bound beads were incubated in the presence or absence of 306 µg of purified bovine heart complex IV (provided by Dr Tsukihara) for 1.5 h at 4 °C and washed twice with 1 ml of the interaction buffer. Co-precipitated proteins were eluted with SDS-containing buffer and analysed via SDS–PAGE and subsequent Coomassie Brilliant Blue (CBB) staining or western blotting with anti-complex IV subunit IV (Abcam, 20E8C12).

**Quantitative PCR.** Analysis of mitochondrial DNA content was performed by quantitative PCR (qPCR) on a 7500 Fast real-time cycler (Applied Biosystems, Thermo Fisher Scientific) using the EXPRESS SYBR GreenER qPCR Supermix kit (ThermoFisher Scientific). Specific primers for mitochondrial DNA (mtDNA; Fw primer: 5′-CAACCATTCATTCCAGCCTT-3′; Rev primer: 5′-GAAAATTT-TAAATGGCCGCA-3′) were designed as previously reported[25]. The relative levels of mitochondrial DNA were normalized to nuclear DNA using primers for actin (Dm_Act79B, QuantiTect Primer Assay, Qiagen). For reverse transcription (RT)-qPCR analyses, the isolation of total RNA was performed using Sepasol-RNA I Super G (Nacalai Tesque). RT was performed using a SuperScript VILO cDNA Synthesis Kit (ThermoFisher Scientific), and subsequent qPCR was performed using the EXPRESS SYBR GreenER qPCR Supermix kit with the following primers (dCHCHD2 Fw primer: 5′-TCCACTCGTCGCACCGCACCTGTG-3′; dCHCHD2 Rev primer: 5′-ACGGCACTGGGAGGAGCGCTCATG-3′; dMICS1 Fw primer: 5′-GCTCAACATCTTTATCCGCATC-3′; dMICS1 Rev primer: 5′-CACCTTAA-ACCAACTCAGTTCTTC-3′; RP49 Fw primer: 5′-ACTTCATCCGCCACCA-GTCG-3′; RP49 Rev primer: 5′-CGGGTGCGCTTGTTCGATCC-3′). The relative transcript levels of the target genes were normalized against RP49 mRNA levels. Quantification was performed using the comparative Ct method.

**Survival assay and climbing assay.** For the lifespan studies, ~20 adult flies per vial (25 mm diameter × 95 mm height) were maintained at 25 °C, transferred to fresh fly food, and scored for survival every 4 days. To control for isogeny, we used $dCHCHD2^{H43}$ and its revertant for comparison or we crossed $dCHCHD2^{H43}$ and the revertant with driver and transgenic lines that were backcrossed to the $w^-$ background for six generations or generated in the $w^-$ background. For the oxidative stress assays, the survival rate of 3-day-old male adult flies kept in a vial containing tissue paper soaked with 0.5% $H_2O_2$ or 2 mM paraquat prepared in Schneider's insect medium was measured as described[16]. Starvation assay was performed with distilled water instead of toxins. For a climbing assay, vials (25 mm diameter × 180 mm height) containing 20–25 flies were tapped gently on the table and left standing for 18 seconds. The number of flies that climbed at least 60 mm was recorded.

**Exposure to hyperoxia.** Adult males (1–2-day-old) in plastic vials (20–25 flies per vial) containing laboratory-standard food were placed in an atmosphere chamber at room temperature supplied with 90% $O_2$ balanced by nitrogen. Each experiment was conducted on at least 50 flies of a given genotype. Flies were removed after 4 days for EM studies and western blot analysis.

**Statistical analysis.** Two-tailed student's t-test or a one-way repeated measures analysis of variance was used to determine significant differences between two or among multiple groups, respectively, unless otherwise indicated. If a significant result was determined using analysis of variance ($P < 0.05$), the mean values of the control and the specific test group were analysed using a Tukey–Kramer test. Dunnett's test were used to determine significant differences between two specific groups or among multiple groups of interests. Data distribution was assumed to be normal, but this was not formally tested. With the exception of collecting data for Drosophila mitochondrial phenotypes, data collection and analysis were not performed blind to the conditions of the experiments and no randomization was used.

**Data availability.** The data that support the findings of this study are available from the corresponding author upon reasonable request.

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

## Acknowledgements

We thank Drs S.H.Y. Loh, L.M. Martins, A. Sánchez-Martínez, H. Tricoire, T. Tsukihara, K. Shinzawa-Itoh, K. Tanaka, M. Miura and T. Oka. for providing materials. Also we thank T. Arano, T. Kanao, C. Cui, S. Kakuta, M. Yoshida, T. Sakanishi, A. Koyanagi and T. Imura for their technical assistance. This study was funded by the Grant-in-Aid for Scientific Research (26293070 (Y.I.), 15H04842 (N.H.)) from MEXT in Japan, the Grant-in-Aid for Scientific Research on Innovative Areas (23111003 (N.H.)), and was partly supported by the 'Platform for Drug Discovery, Informatics, and Structural Life Science' grant from the Japan Agency for Medical Research and Development (AMED, M.U. and J.T.) and by a grant from Otsuka Pharmaceutical (N.H. and Y.I.).

## Author contributions

Conceptualization was done by Y.I. and N.H. Methodology was provided by H.M., C.Y., K.S.-F., T.I., T.N., J.T and Y.I. Investigation was done by H.M., C.Y., K.S.-F., T.H. and M.U. Writing the original draft and visualization were done by H.M., C.Y., M.U., J.T. and Y.I. Review and editing were Y.I. and N.H. Funding acquisition was done by Y.I. and N.H. Resources were acquired by M.F. and S.S. Supervision was done by Y.I. and N.H.

## Additional information

**Competing interests:** The authors declare no competing financial interests.

