## [Peer Review File · Nature Communications]

Reviewers' comments:

Reviewer #1 (Remarks to the Author):

Meng et al investigated the loss of CHCHD2 function on oxidative stress, mitochondrial function and apoptosis in *Drosophila* and mammalian cell lines. Mutations in CHCHD2 have been reported to cause a very rare form of familial Parkinson disease (PD), while some cohorts report no association between CHCHD2 mutations and PD. Flies lacking CHCHD2 exhibited some mitochondrial dysfunction, evidence of increased oxidative stress and modest loss of dopaminergic neuron loss: features typically associated with PD pathogenesis. The authors go on to show that wild type CHCHD2 interacts with MICS1 and cytochrome c and might be involved in regulation of mitochondrial mediated-apoptosis.

Not too much is reported about CHCHD2 and mitochondrial function in the literature and the interaction between this protein and MICS1 to my knowledge is a novel finding. Loss of CHCHD2 could well result in mitochondrial-mediated apoptosis and account for some of the impairments found in flies (e.g. climbing ability, dopaminergic cell loss).

Overall, it is quite a comprehensive study and generally well executed. However they place great emphasis on oxidative stress being a central cog in the pathogenesis. I feel that further experiments are required to confirm such a claim (see below for more). For example, the authors claim 4E-BP is induced in their CHCHD2 deficient flies as a consequence of oxidative stress. True they show that over expression of 4E-BP can improve survival/climbing ability (Fig 7). However they do not show that markers of oxidative stress are reduced in these models, which would strengthen their hypothesis.

Other specific points:

Figure 1: the scoring system for cristae morphology should be moved to the main figure rather than being tucked away in the supplementary. This is one of the central findings so readers shouldn't have to go to the supplementary to understand what class I and class II means in Fig 1e.

Line 124: The authors claim dCHCHD2 cause chronic oxidative stress. However, at this point the authors have only looked at Dj1b and SOD1, which were unchanged. Therefore they cannot conclude this as yet. They go on to show a bit of lipid and DNA oxidation by immunofluorescence. Further to above, more markers of oxidative stress would be useful (glutathione levels, oxidised proteins by OXYblot/ELISA). In particular measuring superoxide production by mitochondria using MitoSOX would be useful in the embryonic *drosophila* cells/MEFS.

Line 188: Following no changes in the activities of the individual mitochondrial complexes (Fig S4) the authors conclude that proper electron flow is maintained in the absence of CHCHD2. They cannot conclude this from the assays as they rely on the addition of exogenous electron carriers (ubiquinone, cytochrome c). Surely the disrupted cristae and seahorse experiments indicate that electron flow is most likely perturbed?

Figure 5: Data for interaction between MICS1 and cytochrome c should be shown. Currently listed as data not shown. Similarly it is mentioned in passing that mass spectrometry indicated interaction between CHCHD2 and MIC1, yet no further data is included in the paper. This should also be included.

In Fig 5d increased cytochrome c is found in the cytosol after treatment as expected with authors implying release from mitochondria. However the mitochondrial pool of cytochrome c does not go down. How do the authors explain this discrepancy? A western should be included to show that WT and mutant forms of CHCHD2 have similar expression levels in MEFs treated with lentivirus. Also, what was the transfection efficiency? The measurement of caspase activity is tucked away in Fig

S6. This data nicely confirms the link with apoptosis and should be moved to the main figure

Figure 6e: data should be in the same figure and stats performed across all groups e.g. LacZ vs. hCHCHD2 WT.

Figure 7A: I don't understand why this blot is included. Authors don't explain it in the text either.

Discussion: In line 301, authors propose complex IV activity is compromised. They do not have data to support this so should be removed.

Minor points:

Line 288: it would be useful to mention in the text the % of MICS KD

Figure 6a: Not really mentioned in text. Should say expression of WT and mutant similar in *Drosophila* lines.

Fig S7 is only referred to in the discussion. The data doesn't really show anything/convincing, so if the authors (like me) don't feel it is necessary to mention it in the results section it should be removed completely.

Reviewer #2 (Remarks to the Author):

In this study Meng and colleagues describe a mechanism by which the mitochondrial protein CHCHD2, whose mutations have recently been reported to cause autosomal dominant Parkinson's disease (PD), induces mitochondrial dysfunction and neurodegeneration. Using *Drosophila* and cultured mammalian cells, the authors report here that the loss of CHCHD2 function would lead to abnormal mitochondrial structure / function, increased oxidative stress, and ultimately neurodegeneration and motor impairment in a time dependent manner. Mechanistically, the study demonstrates that CHCHD2 regulates oxidative phosphorylation by interacting with cytochrome c and MICS1. Overall, this is a very well-done study. Appropriate techniques and controls were used. Conclusions are supported by results. The manuscript is well-written and data are clearly presented. This study should be of interest to the researchers in the field because first, it provides new insights into the pathogenic mechanism of CHCHD2. Second, it reaffirms the role of mitochondrial dysfunction in PD, a topic that has increasingly gained attention in the field.

Reviewer #3 (Remarks to the Author):

In this paper, the authors generated loss-of-dCHCHD2 flies as a new *Drosophila* PD model. This provides a *Drosophila* model to study the in vivo function of the PD-related gene CHCHD2 under physiological and pathological conditions. The authors also found that dCHCHD2 regulates mitochondrial functions through regulation of Cyt c and cristae integrity together with MICS1, and that 4E-BP over-expression suppresses PD-related phenotypes in dCHCHD2 mutant flies. These findings provide insights into the mitochondrial pathogenesis of PD. In this paper, data on characterization of dCHCHD2 mutant phenotypes are relative clear; however, data on the genetic interactions between dCHCHD2 and MICS1, between dCHCHD2 and 4E-BP are relatively weak and need to be strengthened.

Some general comments:

The authors suggest that oxidative stress and cell death signaling might be the cause of PD pathology; however, they have not supplied enough evidence to substantiate their claims. To directly visualize the production of ROS, they could perhaps perform DHE staining. Suppression of the mutant phenotypes by 4EBP overexpression alone is not sufficient to claim that these phenotypes are due to oxidative stress as 4EBP also plays roles other than alleviating oxidative stress, especially in light of the fact that DJ-1 overexpression, which is anti-oxidative, did not

rescue the phenotypes in their hands. To confirm that oxidative stress is indeed causing the phenotypes, they can test overexpression of SOD1.

Although there seems to be loss of TH⁺ neurons, it is unclear whether it is really due to apoptotic cell death. To test this, they should perform TUNEL assays both in the brain and in the indirect flight muscles of the dCHCHD2 mutant.

Interestingly, although dCHCHD2 is upregulated in PINK1/parkin mutant flies, there was no evidence that dCHCHD2 mutant exacerbates PINK1/parkin mitochondrial phenotypes. It might still be interesting to see if overexpression of PINK1 or parkin can suppress the phenotypes of the dCHCHD2 mutant.

Finally, the authors make the general comment regarding their work that

" Here, we report

78 that the loss of CHCHD2 in flies leads to mitochondrial and neuronal phenotypes
79 associated with PD pathology, including increased sensitivity to oxidative stress and
80 loss of dopaminergic (DA) neurons with age. These phenotypes were rescued by 4E-BP
81 and human CHCHD2 but not by CHCHD2 mutants found in PD cases. Our study
82 suggests that CHCHD2 has both gain- and loss-of-function aspects in PD, exacerbating
83 oxidative stress and cell death signaling."

From this they want to conclude that CHCHD2 has gain and loss of function phenotypes in PD. But, I am not sure if this statement can be made. Just because expression of a CHCHD2 mutant does not rescue does not mean it is loss of function. It could also be a neomorph, with novel activities that cause disease. I think the author's statement below regarding dimer formation is consistent with this possibility.

"that the 252 T61I mutation, which occurs in a central conserved
253 region (Fig. 1a), destabilized the CHCHD2 protein, leading to its appearance in the
254 insoluble fraction when expressed in bacteria (Supplementary Fig. 6d). CHCHD2
255 formed a homodimer (Supplementary Fig. 6e). R145Q, which is located adjacent to the
256 second Cx9C motif (Fig. 1a), affected the dimer formation (Supplementary Fig. 6e)."

In short, I do not think that key experiments have been done. One key experiment is whether heterozygosity for a disease mutant causes mitochondrial phenotypes that are different from those associated with heterozygosity for a null mutation. This would address the question as to whether the mutant form has dominant activity. If phenotypes are observed it would then be important to know if overexpression of the wild type can suppress these, which would be consistent with dominant effects on the normal activity of the protein.

More detailed figure-by-figure reviews are as follows:

Figure 1c: Besides formation of mitochondrial swirls, dCHCHD2 mutants also seem to show altered mitochondrial size and irregular mitochondrial shape. Also, given that over-expression of MICS1 results in mitochondrial fragmentation (as shown in Figure 5f), does over-expression of dCHCHD2 result in fragmented mitochondria as well? - So the question is if dCHCHD2 also regulates mitochondrial morphology/dynamics, an important aspect in the regulation of mitochondrial integrity. It would be important if the authors also quantify mitochondrial size in the dCHCHD2 mutant and over-expression flies as compared to wild-type flies, and show representative EM images.

Figure 1c-f: Given the severe mitochondrial cristae defects and reduced ATP levels in dCHCHD2 mutants shown in Figure 1, do dCHCHD2 mutants show apoptotic muscle death (TUNEL positivity)? This is an especially interesting question since the authors also show in Figure 5 that dCHCHD2 suppresses Cyt c release as well as apoptosis. It would also be good if the authors can assay tissue

integrity in dCHCHD2 mutants by Toluidine Blue staining of plastic sections.

Figure 3c-d: Previous research reported that hyperoxia induces mitochondrial swirls in wild type flies (Walker and Benzer, 2004), reminiscent of what the authors observed in aged dCHCHD2 mutant flies in Figure 1. Did the authors observe the same phenotype under hyperoxia treatment here? Did dCHCHD2H43 / dCHCHD2^{-/-} under hyperoxia condition show more swirls than under normoxia? If so representative EM images should be provided in addition to the statistics in Figure 3d. It is also important to note that mitochondrial swirls are not generally acknowledged as reflecting oxidative stress. While the Benzer paper showed that hyperoxia induces mitochondrial swirls this does not mean (mechanistically, since we do not know how swirls are formed) that any oxidative stress leads to swirls. This is something that would need to be more generally established in order for the phenotype to be used as a general marker for such a phenotype.

Figure 5b: MICS1 is upregulated gradually under low serum conditions for 24-72 hours. However, why do CHCHD2 levels drop back after 72 hours of low serum conditions, after an upregulation at 48-hour time-point?

Figure 5e-h: Genetic interactions between dCHCHD2 and MISC1 should be strengthened – since MICS1 over-expression rescues the cristae defects, mitochondrial swirls and ATP reduction in dCHCHD2 mutants (Figure 5f-h), can MICS1 over-expression also rescue shortened lifespan, climbing defects and DA neuron loss in dCHCHD2 mutants?

Figure 6b: It is important to show more representative images with mitochondrial swirls in the LacZ and hCHCHD2R145Q panels.

Figure 6e: For each row, left and right panels should be combined into one, and each genotype should be compared to LacZ to calculate statistical significance.

Figure 7c-e: Genetic interactions between dCHCHD2 and 4E-BP should be strengthened – does 4E-BP over-expression also rescue reduced ATP levels, and more importantly, the cristae defects and mitochondrial swirls in dCHCHD2 mutant flies? It would also be great to show if all the dCHCHD2 mutant phenotypes can be suppressed by Rapamycin treatment (activation of 4E-BP) (Tain et al., 2009).

Point-by-point responses to reviewer comments

We greatly appreciate the feedback from the three reviewers on our manuscript (NCOMMS-16-12842) entitled “**The loss of Parkinson’s disease-associated protein CHCHD2 affects mitochondrial crista structure and destabilizes cytochrome c**”. We found the comments to be highly constructive for improving our manuscript. We have now performed additional experiments to address the questions and concerns of the reviewers, and we have incorporated the new results into the revised manuscript. Below are our point-by-point responses to the reviewers’ comments. Original comments by the Reviewers are shown in blue. Papers cited here correspond to a list of **References for responses** shown after the responses.

Reviewer #1:

General comments:

Overall, it is quite a comprehensive study and generally well executed. However they place great emphasis on oxidative stress being a central cog in the pathogenesis. I feel that further experiments are required to confirm such a claim (see below for more). For example, the authors claim 4E-BP is induced in their CHCHD2 deficient flies as a consequence of oxidative stress. True they show that over expression of 4E-BP can improve survival/climbing ability (Fig 7). However they do not show that markers of oxidative stress are reduced in these models, which would strengthen their hypothesis.

In response to the reviewer’s suggestion, we confirmed that an oxidative stress marker was reduced by 4E-BP (Fig S7a). However, our data suggesting that SOD1 and DJ-1 do not modulate *dCHCHD2* phenotypes strongly imply that oxidative stress is only part of the pathogenesis caused by CHCHD2 mutations. Thus, we do not think it is appropriate to emphasize an anti-oxidant effect of 4E-BP in this manuscript. Some studies have demonstrated that 4E-BP has multiple beneficial roles in mitochondrial homeostasis, which includes modulation of OXPHOS functions (Morita *et al.*, 2013; Zid *et al.*, 2009). We discuss this point and modify our conclusions about the possible roles of 4E-BP in *CHCHD2* mutant flies. Details will be explained in the point-by-point responses.

Other specific points:

Figure 1: the scoring system for cristae morphology should be moved to the main figure rather than being tucked away in the supplementary. This is one of the central findings so readers shouldn't have to go to the supplementary to understand what class I and class II means in Fig 1e.

We moved the figure for the scoring system to the main figure (it is now Fig 1d).

Line 124: The authors claim *dCHCHD2* cause chronic oxidative stress. However, at this point the authors have only looked at *Dj1b* and *SOD1*, which were unchanged. Therefore they cannot conclude this as yet. They go on to show a bit of lipid and DNA oxidation by immunofluorescence. Further to above, more markers of oxidative stress would be useful (glutathione levels, oxidised proteins by OXYblot/ELISA). In particular measuring superoxide production by mitochondria using MitoSOX would be useful in the embryonic drosophila cells/MEFS.

In response to the reviewer's suggestions, we conducted several sets of experiments to examine the possibility that *CHCHD2* loss causes chronic oxidative stress.

We first estimated superoxide production in embryonic cells using MitoSox, which indicated that loss of *CHCHD2* resulted in increased superoxide production (fold increase: 1.77 +/- 0.24 for average fluorescence intensity from three independent experiments). Data were inserted as Fig 2e, f.

We next measured the GSH/GSSG ratios of flies, which decline as the flies age. This result clearly indicates that loss of *dCHCHD2* accelerates GSH oxidation at a younger age (i.e., 22 days old). These data were inserted as Fig S2b. An oxyblot was also carried out in the thoraxes of 14-day-old and 30-day-old flies. Although we were not able to detect significant differences in 14-day-old flies (data not shown), loss of *dCHCHD2* promoted protein oxidization by 30 days old, suggesting that chronic oxidative stress occurs in *dCHCHD2* mutant flies (Fig S2a). Finally, we tested whether

SOD1 or DJ-1 overexpression rescues the *dCHCHD2* phenotypes (Fig S7c-f). SOD1 and DJ-1 failed to rescue the mutant phenotypes, which marks a sharp contrast with 4E-BP. We discuss this issue as follows in the Discussion section:

Although multiple lines of evidence indicate that chronic oxidative stress occurs by CHCHD2 loss, overexpression of SOD1 and DJ-1 or removal of DJ-1 did not affect dCHCHD2 phenotypes, which strongly suggests that oxidative stress is only part of the pathogenesis of PD that is linked to CHCHD2. The functional difference between SOD1/DJ-1 and 4E-BP is that the latter regulates mitochondrial functions and proteostasis in addition to upregulation of anti-oxidative proteins such as anti-oxidant GST-S1 (Demontis and Perrimon, 2010; Morita et al., 2013; Tain et al., 2009; Zid et al., 2009). For mitochondrial regulation by 4E-BP, it has been demonstrated that 4E-BP is involved in translational upregulation of complex I and IV subunits under dietary restriction, which enhances OXPHOS (Zid et al., 2009). Another study has proposed that 4E-BP suppresses OXPHOS proteins and mitochondrial respiration (Morita et al., 2013). We favor the latter idea because hypoxia improves the mitochondrial defects of dCHCHD2 flies (Supplementary Fig. 8e,f).

Line 188: Following no changes in the activities of the individual mitochondrial complexes (Fig S4) the authors conclude that proper electron flow is maintained in the absence of CHCHD2. They cannot conclude this from the assays as they rely on the addition of exogenous electron carriers (ubiquinone, cytochrome c). Surely the disrupted cristae and Seahorse experiments indicate that electron flow is most likely perturbed?

In Fig S4, we measured the activities of OXPHOS complexes (I-IV) but not electron flow between the complexes. Although the enzyme activities of complexes I-IV were not significantly altered (Fig S4a), OCR was impaired in the *CHCHD2*-deficient flies after FCCP treatment, suggesting that the electron flow from complexes I and II to IV was at least partially compromised (Fig 4). Please also refer to the below figure from a paper explaining the principle of OCR measurement (Pelletier *et al.*, 2014).

CHCHD2 binds to Cyt c (Fig 5a,b), and its absence facilitates Cyt c release after actinomycin D and peroxide treatment in cultured cells (Fig 5c,d and new Fig S5f). Based on these observations, we hypothesize that CHCHD2 and MICS1 stabilize Cyt c to maintain normal electron flow during OXPHOS (Fig 7h) although, due to technical limitations, we do not show direct data establishing that the electron flow from complex III to complex IV is regulated by CHCHD2.

Figure legend. FCCP is an uncoupling agent that permeabilizes the inner mitochondrial membrane to protons, forcing the mitochondria to increase the flow of electrons (and thus oxygen consumption) to maintain the membrane potential. Block arrows represent the electron flow in the mitochondrial electron transport chain. The two parameters measured with an extracellular flux analyzer, namely the rate of oxygen (O_2) consumption and the release of protons (H^+) during lactate production, are depicted in boxes.

Figure 5: Data for interaction between MICS1 and cytochrome c should be shown. Currently listed as data not shown.

We added data showing endogenous interactions among MICS1, Cyt C and CHCHD2 (Fig 5b), which should satisfy the reviewer's request and strengthen our manuscript.

Similarly it is mentioned in passing that mass spectrometry indicated interaction between CHCHD2 and MIC1, yet no further data is included in the paper. This should also be included.

We added the mass spectrometry data to Fig S5a.

In Fig 5d increased cytochrome c is found in the cytosol after treatment as expected with authors implying release from mitochondria. However the mitochondrial pool of cytochrome c does not go down. How do the authors explain this discrepancy?

In additional experiments (shown in Fig 5d and Fig S5f), we modified the number of strokes used for homogenization (20 passages through a 25-G syringe) and reduced the amounts of mitochondrial fractions to be subjected to western blot analysis to avoid saturating the mitochondrial Cyt c signals. The blot images and quantitative data were replaced with the results from the new experiments.

A western should be included to show that WT and mutant forms of CHCHD2 have similar expression levels in MEFs treated with lentivirus. Also, what was the transfection efficiency?

We added data showing similar expression levels in CHCHD2 WT and mutants in Fig S5e. CHCHD2 genes were chromosomally integrated into MEFs by using a retroviral vector, pMXs-puro, and transfected cells were selected by 2 μ g/ml puromycin for a few

days. Under this condition, 100% of surviving cells expressed the transgenes. This result was evaluated by immunostaining, as shown in the images below:

The measurement of caspase activity is tucked away in Fig S6. This data nicely confirms the link with apoptosis and should be moved to the main figure.

We moved the caspase data to Fig 5e, f.

Figure 6e: data should be in the same figure and stats performed across all groups e.g. LacZ vs. hCHCHD2 WT.

Because the differences between hCHCHD2 WT and the pathogenic mutant lines were too small to detect significant differences compared to the LacZ group, we repeated these experiments. The new data were combined with the previous data.

Figure 7A: I don't understand why this blot is included. Authors don't explain it in the text either.

We observed a slight upregulation of 4E-BP when we overexpressed β -galactosidase (LacZ) and CHCHD2 proteins by using the *MHC-GAL4* driver. This result might be due to protein overexpression and/or GAL4 overproduction rather than oxidative stress. 4E-BP is upregulated by ATF4 (Yamaguchi *et al.*, 2008). ATF4, which is conserved between mammals and *Drosophila*, is activated by unfolded protein stress and by some types of mitochondrial stress (Munch and Harper, 2016). To overcome this problem, we used the inducible GAL4 system (GeneSwitch-GAL4) to achieve endogenous levels of CHCHD2 and titrated the amounts of a GAL4 inducer, RU486, to induce CHCHD2 to endogenous levels (Data were moved to Fig S6g). We explained the meaning of Fig. S6g briefly in the text as follows;

Overexpressing CHCHD2 and LacZ with the GAL4-UAS system caused 4E-BP expression to increase slightly, probably due to GAL4, LacZ and/or CHCHD2 overexpression. However, reintroduction of dCHCHD2 at a level similar to endogenous levels suppressed 4E-BP induction in dCHCHD2-deficient flies (Fig. 7a, Supplementary Fig. 6g).

Discussion: In line 301, authors propose complex IV activity is compromised. They do not have data to support this so should be removed.

Reference 31 reports that the activity of complex IV declines with age in flies, but complex IV protein levels do not decline. Thus, we would like to discuss the possibility of age-dependent decline of complex IV activity and Cyt c dysregulation due to

CHCHD2 mutations additively or synergistically enhancing ROS production in aged *dCHCHD2*-deficient flies. As the previous description was speculative, we modified the sentence as follows:

This observation raises the possibility that the function of complex IV, in addition to that of Cyt c, is compromised in dCHCHD2-deficient flies as they aged, resulting in a high incidence of reactive oxygen species.

Minor points:

Line 288: it would be useful to mention in the text the % of MICS KD.

We mentioned the KD efficiency in the text.

Figure 6a: Not really mentioned in text. Should say expression of WT and mutant similar in *Drosophila* lines.

We mentioned the expression of *hCHCHD2* as follows;

We introduced LacZ, dCHCHD2 or hCHCHD2, including WT and two mutants, into dCHCHD2^{H43} flies; hCHCHD2 WT and mutant genes were expressed at similar levels (Fig. 6a).

Fig S7 is only referred to in the discussion. The data doesn't really show anything/convincing, so if the authors (like me) don't feel it is necessary to mention it in the results section it should be removed completely.

We removed the data for autophagy and mitophagy (Fig S7c-e) as they are not necessary in this manuscript. However, data from the genetic tests with DJ-1, PINK1 and Parkin were kept because these data should be of interest to most researchers in this field. We mentioned these results in the Results section as the reviewer suggested.

Reviewer #2:

In this study Meng and colleagues describe a mechanism by which the mitochondrial protein CHCHD2, whose mutations have recently been reported to cause autosomal dominant Parkinson's disease (PD), induces mitochondrial dysfunction and neurodegeneration. Using *Drosophila* and cultured mammalian cells, the authors report here that the loss of CHCHD2 function would lead to abnormal mitochondrial structure / function, increased oxidative stress, and ultimately neurodegeneration and motor impairment in a time dependent manner. Mechanistically, the study demonstrates that CHCHD2 regulates oxidative phosphorylation by interacting with cytochrome c and MICS1. Overall, this is a very well-done study. Appropriate techniques and controls were used. Conclusions are supported by results. The manuscript is well-written and data are clearly presented. This study should be of interest to the researchers in the field because first, it provides new insights into the pathogenic mechanism of CHCHD2. Second, it reaffirms the role of mitochondrial dysfunction in PD, a topic that has increasingly gained attention in the field.

We are pleased that Reviewer #2 evaluates our manuscript by the description “This study should be of interest to the researchers in the field because first, it provides new insights into the pathogenic mechanism of CHCHD2. Second, it reaffirms the role of mitochondrial dysfunction in PD, a topic that has increasingly gained attention in the field”.

Reviewer #3:

In this paper, data on characterization of dCHCHD2 mutant phenotypes are relative clear; however, data on the genetic interactions between dCHCHD2 and MICS1, between dCHCHD2 and 4E-BP are relatively weak and need to be strengthened.

We added additional data on MICS1 and 4E-BP to strength the manuscript. We will response these issues in the point-by-point responses.

Some general comments:

The authors suggest that oxidative stress and cell death signaling might be the cause of PD pathology; however, they have not supplied enough evidence to substantiate their claims. To directly visualize the production of ROS, they could perhaps perform DHE staining. Suppression of the mutant phenotypes by 4EBP overexpression alone is not sufficient to claim that these phenotypes are due to oxidative stress as 4EBP also plays roles other than alleviating oxidative stress, especially in light of the fact that DJ-1 overexpression, which is anti-oxidative, did not rescue the phenotypes in their hands. To confirm that oxidative stress is indeed causing the phenotypes, they can test overexpression of SOD1.

Although our new data [MitoSox (Fig 2e,f) and GSH/GSSG (Fig S2b) measurements and Oxyblot (Fig S2a)] reinforced the possibility that CHCHD2 loss causes chronic oxidative stress from mitochondria, SOD1 and DJ-1 overexpression failed to rescue it (new Fig S7c-f), which marks a sharp contrast with 4E-BP. We discuss this issue as follows in the Discussion section:

Although multiple lines of evidence indicate that chronic oxidative stress occurs by CHCHD2 loss, overexpression of SOD1 and DJ-1 or removal of DJ-1 did not affect dCHCHD2 phenotypes, which strongly suggests that oxidative stress is only part of the pathogenesis of PD that is linked to CHCHD2. The functional difference between SOD1/DJ-1 and 4E-BP is that the latter regulates mitochondrial functions and proteostasis in addition to upregulation of anti-oxidative proteins such as anti-oxidant GST-S1 (Demontis and Perrimon, 2010; Morita et al., 2013; Tain et al., 2009; Zid et al., 2009). For mitochondrial regulation by 4E-BP, it has been demonstrated that 4E-BP is involved in translational upregulation of complex I and IV subunits under dietary restriction, which enhances OXPHOS (Zid et al., 2009). Another study has proposed that 4E-BP suppresses OXPHOS proteins and mitochondrial respiration (Morita et al.,

2013). We favor the latter idea because hypoxia improves the mitochondrial defects of *dCHCHD2* flies (Supplementary Fig. 8e,f).

Although there seems to be loss of TH+ neurons, it is unclear whether it is really due to apoptotic cell death. To test this, they should perform TUNEL assays both in the brain and in the indirect flight muscles of the *dCHCHD2* mutant.

We performed TUNEL assays in both the brain (Fig S1h) and the indirect flight muscles (Fig 1g). Although we detected significant TUNEL-positive muscle cells in 30-day-old *CHCHD2*-deficient flies, there was no difference in the number of TUNEL-positive cells in the brain tissues of 45-day-old flies. Given that a few TH-positive neurons are lost in 50-day-old flies, the result is not surprising. By contrast, we observed a significant reduction in anti-TH signal intensity, suggesting that most of the TH-positive neurons suffer from subclinical stress (Fig S1h).

Interestingly, although *dCHCHD2* is upregulated in *PINK1/parkin* mutant flies, there was no evidence that *dCHCHD2* mutant exacerbates *PINK1/parkin* mitochondrial phenotypes. It might still be interesting to see if overexpression of *PINK1* or *parkin* can suppress the phenotypes of the *dCHCHD2* mutant.

We overexpressed *Drosophila* *PINK1* and *Parkin* in *dCHCHD2*-deficient flies (Fig S8c, d). Overexpression of either *PINK1* or *Parkin* promoted mitochondrial fragmentation, but the integrity of the crista was maintained as reported. Surprisingly, their overexpression apparently exacerbated the mitochondrial dysfunction caused by the loss of *dCHCHD2*. We discuss this result in the Discussion as follows:

In sharp contrast to 4E-BP overexpression, PINK1 or Parkin overexpression decreased mitochondrial integrity of dCHCHD2-deficient flies. According to the hypothesis that PINK1-Parkin signaling is responsible for mitochondrial quality control, PINK1 or Parkin overexpression might activate mitophagy. This effect would cause most of the

mitochondria in dCHCHD2-deficient flies to be recognized as damaged mitochondria, leading to lysosomal degradation.

They want to conclude that CHCHD2 has gain and loss of function phenotypes in PD. But, I am not sure if this statement can be made. Just because expression of a CHCHD2 mutant does not rescue does not mean it is loss of function. It could also be a neomorph, with novel activities that cause disease. I think the author's statement below regarding dimer formation is consistent with this possibility.

"that the 252 T61I mutation, which occurs in a central conserved region (Fig. 1a), destabilized the CHCHD2 protein, leading to its appearance in the insoluble fraction when expressed in bacteria (Supplementary Fig. 6d). CHCHD2 formed a homodimer (Supplementary Fig. 6e). R145Q, which is located adjacent to the second Cx9C motif (Fig. 1a), affected the dimer formation (Supplementary Fig. 6e)."

In short, I do not think that key experiments have been done. One key experiment is whether heterozygosity for a disease mutant causes mitochondrial phenotypes that are different from those associated with heterozygosity for a null mutation. This would address the question as to whether the mutant form has dominant activity. If phenotypes are observed it would then be important to know if overexpression of the wild type can suppress these, which would be consistent with dominant effects on the normal activity of the protein.

We thank Reviewer #3 for these thoughtful comment and suggestions. We agree that this issue is very important to understand the etiology of Parkinson's disease linked to *CHCHD2* mutations. We analyzed the effects of hCHCHD2 mutants (T61I and R145Q) in the *dCHCHD2*^{+/-} genetic background. *dCHCHD2*^{+/-} flies exhibit almost normal mitochondrial phenotypes. If hCHCHD2 mutants confer gain of toxic functions, the introduction of these in *dCHCHD2*^{+/-} flies should produce mitochondrial defects observed in *dCHCHD2*^{-/-} flies expressing hCHCHD2 mutants, as shown in Fig 6b,c,e. Both TEM analysis (Fig S6e,f) and an ATP assay (Fig S6g) indicated that the

introduction of hCHCHD2 mutants does not lead to mitochondrial defects, supporting our idea that disease-associated mutations have a property similar to loss-of-function.

More detailed figure-by-figure reviews:

Figure 1c: Besides formation of mitochondrial swirls, dCHCHD2 mutants also seem to show altered mitochondrial size and irregular mitochondrial shape. Also, given that over-expression of MICS1 results in mitochondrial fragmentation (as shown in Figure 5f), does over-expression of dCHCHD2 result in fragmented mitochondria as well? - So the question is if dCHCHD2 also regulates mitochondrial morphology/dynamics, an important aspect in the regulation of mitochondrial integrity. It would be important if the authors also quantify mitochondrial size in the dCHCHD2 mutant and over-expression flies as compared to wild-type flies, and show representative EM images.

We measured the muscle mitochondrial length in the direction of the long axis by using mitoGFP signals to process many independent samples. The average mitochondrial lengths of *dCHCHD2*-deficient flies expressing dCHCHD2 WT, human CHCHD2 WT or PD mutants were not significantly different from those of *dCHCHD2*-deficient flies expressing a control *LacZ* transgene, suggesting that CHCHD2 does not directly regulate mitochondrial morphology and dynamics (Dunnett's test *vs.* *LacZ*, $n = 81$ in 9 different images from 3 independent flies.). The graph and representative fluorescence images were added as Fig S8b.

Figure 1c-f: Given the severe mitochondrial cristae defects and reduced ATP levels in dCHCHD2 mutants shown in Figure 1, do dCHCHD2 mutants show apoptotic muscle death (TUNEL positivity)? This is an especially interesting question since the authors also show in Figure 5 that dCHCHD2 suppresses Cyt c release as well as apoptosis. It would also be good if the authors can assay tissue integrity in dCHCHD2 mutants by Toluidine Blue staining of plastic sections.

As mentioned above, we performed a TUNEL assay in the muscle sections (Fig 1g). Additionally, we performed toluidine blue staining on serial sections used in the TUNEL assay (Fig 1g). In the 30-day-old *CHCHD2*-deficient flies, muscle integrity was reduced, and there was mild muscle fiber atrophy.

Figure 3c-d: Previous research reported that hyperoxia induces mitochondrial swirls in wild type flies (Walker and Benzer, 2004), reminiscent of what the authors observed in aged *dCHCHD2* mutant flies in Figure 1. Did the authors observe the same phenotype under hyperoxia treatment here? Did *dCHCHD2^{H43} / dCHCHD2^{-/-}* under hyperoxia condition show more swirls than under normoxia? If so representative EM images should be provided in addition to the statistics in Figure 3d. It is also important to note that mitochondrial swirls are not generally acknowledged as reflecting oxidative stress. While the Benzer paper showed that hyperoxia induces mitochondrial swirls this does not mean (mechanistically, since we do not know how swirls are formed) that any oxidative stress leads to swirls. This is something that would need to be more generally established in order for the phenotype to be used as a general marker for such a phenotype.

We added EM images of mitochondria exhibiting ‘swirls’ under hyperoxia treatment to Fig 3c, and we placed a graph of swirl frequency under normoxic and hyperoxic conditions in Fig 3d. Because we used very young adult flies (5-6 days old in sampling time) in that assay, we did not see a significant difference in the number of ‘swirl’ mitochondria between *dCHCHD2^{+/+}* and *dCHCHD2^{-/-}* flies under normoxia. However, the number of ‘swirl’ mitochondria was significantly increased under hyperoxia treatment for 4 days, and this difference was exacerbated in the absence of *CHCHD2* ($p < 0.05$ by Tukey–Kramer test).

Our hyperoxia-induced swirl frequency in the control *w¹¹¹⁸* flies (10%) was lower than a result (35%) previously reported in *PNAS* (Walker and Benzer, 2004), which might be due to a difference in experimental conditions (90% O₂ for 4 days vs. 100% O₂ for 4 days). However, we frequently observed mitochondria with disordered crista arrays, even in *w¹¹¹⁸* flies, as shown in the upper left image of Fig 3c. These mitochondria

might represent ‘pre-swirl’ structures. Thus, we would like to use the previous two images (upper left and middle in Fig 3c) as representative observations.

Inhibition of QIL1 (C19orf70), a protein that regulates the cristae junctions (CJs), has been reported to exhibit ‘swirl mitochondria’ in *Drosophila* (Guarani *et al.*, 2015). Because crista structure maintains the OXPHOS complex, inhibition of QIL1 leads to reduced respiration. Recent studies, including this work, have proposed that the formation of CJs is a dynamic process; it may respond to metabolic or hyperoxic states and may be impaired with aging. ‘Swirl mitochondria’ might be a marker of dysregulation of cristae and CJ integrity, although further studies are required to fully understand how swirls are formed. We modified a sentence in the Results section as follows:

Hyperoxia induces the mitochondrial swirl phenotype and defects in the Cyt c oxidase complex, which is likely to be accompanied by oxidative stress and/or dysregulation of the cristae junctions (Guarani et al., 2015; Walker and Benzer, 2004).

Figure 5b: MICS1 is upregulated gradually under low serum conditions for 24-72 hours. However, why do CHCHD2 levels drop back after 72 hours of low serum conditions, after an upregulation at 48-hour time-point?

We tested the possible involvement of HtrA2/Omi in CHCHD2 reduction and could not find any evidence that HtrA2/Omi processes CHCHD2 (see Figures below). We also treated cells with several inhibitors, including a pan-caspase inhibitor, Z-VAD-fmk (100 μ M), a proteasome inhibitor, MG-132 (0.5 and 5 μ M), autophagy inhibitors bafilomycin A1 (100 μ M), pepstatin A/E-64d (10 μ g/ml each) at the point of 60 hr (see Figures below). Surprisingly, the inhibition of autophagy protected CHCHD2 reduction. This finding could be potentially interesting. However, we would like to remove the data of Fig 5b for the following reasons:

1) Fig 5b shows only the expression of two proteins, CHCHD2 and MICS1; it does not by itself explain the functional association of the two proteins.

2) The possible CHCHD2 regulation mechanism by autophagy (or mitophagy) is beyond the scope of our current manuscript.

Figure 5e-h: Genetic interactions between dCHCHD2 and MISC1 should be strengthened – since MISC1 over-expression rescues the cristae defects, mitochondrial swirls and ATP reduction in dCHCHD2 mutants (Figure 5f-h), can MISC1 over-expression also rescue shortened lifespan, climbing defects and DA neuron loss in dCHCHD2 mutants?

We found that ubiquitous overexpression of dMISC1 by *Da-GAL4* at 25°C results in death at the 1st instar larval stage. Reduced expression levels (*Da-GAL4* at 22°C or 18°C) caused reduced hatching efficiency (~10%), slowed growth and death at the 2nd instar larval stage (see the Figure below), whereas the loss of dCHCHD2 did not affect the developmental phenotype and hatching efficiency of dMISC1-overexpressing larvae. Due to the strong toxicity of dMISC1 overexpression, we could not examine the effects of dMISC1 overexpression on lifespan and neuronal phenotypes in *dCHCHD2* mutants. Therefore, we estimated only the climbing activity of flies overexpressing dMISC1 in

the muscle tissues, in which dMICS1 overexpression showed a rescue effect (Fig S6a). The developmental phenotype of dMICS1 is mentioned in the legend of Fig S6a.

Figure legend. LacZ or dMICS1 was driven by *Da-GAL4* at 22°C. The body lengths of larvae were measured 6 days post hatching. dMICS1 overexpression (OE) led to reduced growth (***) $p < 0.001$), but dCHCHD2 expression did not affect growth or hatching efficiency. $n = 9-21$. Scale bar = 250 μm .

Figure 6b: It is important to show more representative images with mitochondrial swirls in the LacZ and hCHCHD2R145Q panels.

Although mitochondrial swirls, which are prominent structures, were more frequent in dCHCHD2 mutants, the typical frequency of swirls was at most 2~3% (lower and higher magnification EM images to show the frequency and swirl details, respectively, are inserted below.). The mitochondrial abnormalities in *dCHCHD2* mutant flies mainly consisted of dilated or fuzzy cristae. To avoid any misunderstanding about the mitochondrial phenotypes of mutant flies, we would like to keep the images as they are.

Figure legend. TEM images of the indirect flight muscles of 14-day-old flies with the indicated genotypes. (Lower) Higher magnification images of the boxed regions in the upper panels. White arrowheads indicate swirled cristae. Scale bars = 2 μ m (upper) and 1 μ m (lower), respectively.

Figure 6e: For each row, left and right panels should be combined into one, and each genotype should be compared to *LacZ* to calculate statistical significance.

Because the differences between *hCHCHD2* WT and its pathogenic mutant lines were too small to detect significant differences compared to the *LacZ* group, we repeated these experiments. New data were used in combination with the previous data.

Figure 7c-e: Genetic interactions between *dCHCHD2* and 4E-BP should be strengthened – does 4E-BP over-expression also rescue reduced ATP levels, and more

importantly, the cristae defects and mitochondrial swirls in *dCHCHD2* mutant flies? It would also be great to show if all the *dCHCHD2* mutant phenotypes can be suppressed by Rapamycin treatment (activation of 4E-BP) (Tain *et al.*, 2009).

Based on the reviewer's suggestion, we examined whether 4E-BP rescues *dCHCHD2* mitochondrial phenotypes. Overexpression of 4E-BP significantly improved the cristae defects, including swirls and ATP production. Rapamycin administration also alleviated the ATP decline caused by *dCHCHD2* loss. These data were inserted as Fig 7b-d.

References for responses

Demontis, F., and Perrimon, N. (2010). FOXO/4E-BP signaling in *Drosophila* muscles regulates organism-wide proteostasis during aging. *Cell* *143*, 813-825.

Guarani, V., McNeill, E.M., Paulo, J.A., Huttlin, E.L., Frohlich, F., Gygi, S.P., Van Vactor, D., and Harper, J.W. (2015). QIL1 is a novel mitochondrial protein required for MICOS complex stability and cristae morphology. *Elife* *4*.

Morita, M., Gravel, S.P., Chenard, V., Sikstrom, K., Zheng, L., Alain, T., Gandin, V., Avizonis, D., Arguello, M., Zakaria, C., *et al.* (2013). mTORC1 controls mitochondrial activity and biogenesis through 4E-BP-dependent translational regulation. *Cell metabolism* *18*, 698-711.

Munch, C., and Harper, J.W. (2016). Mitochondrial unfolded protein response controls matrix pre-RNA processing and translation. *Nature* *534*, 710-713.

Pelletier, M., Billingham, L.K., Ramaswamy, M., and Siegel, R.M. (2014). Extracellular flux analysis to monitor glycolytic rates and mitochondrial oxygen consumption. *Methods Enzymol* *542*, 125-149.

Tain, L.S., Mortiboys, H., Tao, R.N., Ziviani, E., Bandmann, O., and Whitworth, A.J. (2009). Rapamycin activation of 4E-BP prevents parkinsonian dopaminergic neuron loss. *Nat Neurosci* *12*, 1129-1135.

Walker, D.W., and Benzer, S. (2004). Mitochondrial "swirls" induced by oxygen stress and in the *Drosophila* mutant hyperswirl. *Proceedings of the National Academy of Sciences of the United States of America* *101*, 10290-10295.

Yamaguchi, S., Ishihara, H., Yamada, T., Tamura, A., Usui, M., Tominaga, R., Munakata, Y., Satake, C., Katagiri, H., Tashiro, F., *et al.* (2008). ATF4-mediated induction of 4E-BP1 contributes to pancreatic beta cell survival under endoplasmic reticulum stress. *Cell metabolism* *7*, 269-276.

Zid, B.M., Rogers, A.N., Katewa, S.D., Vargas, M.A., Kolipinski, M.C., Lu, T.A., Benzer, S., and Kapahi, P. (2009). 4E-BP extends lifespan upon dietary restriction by enhancing mitochondrial activity in *Drosophila*. *Cell* *139*, 149-160.

REVIEWERS' COMMENTS:

Reviewer #1 (Remarks to the Author):

I am happy with all the changes.

Reviewer #2 (Remarks to the Author):

The authors have been responsive to the reviewers' comments. I am satisfied with the revised manuscript.